# Age-Related Alterations at Neuromuscular Junction: Role of Oxidative Stress and Epigenetic Modifications

**DOI:** 10.3390/cells10061307

**Published:** 2021-05-24

**Authors:** Gabriella Dobrowolny, Alessandra Barbiera, Gigliola Sica, Bianca Maria Scicchitano

**Affiliations:** 1Department of Anatomy, Histology, Forensic Medicine and Orthopaedics (DAHFMO)-Unit of Histology and Medical Embryology, Sapienza University of Rome, Laboratory Affiliated to Istituto Pasteur Italia-Fondazione Cenci Bolognetti, 00161 Rome, Italy; Gabriella.Dobrowolny@uniroma1.it; 2Department of Life Sciences and Public Health, Histology and Embryology Unit, Fondazione Policlinico Universitario A. Gemelli IRCCS, 00168 Rome, Italy; Alessandra.Barbiera@unicatt.it (A.B.); Gigliola.Sica@unicatt.it (G.S.)

**Keywords:** aging, ROS, oxidative stress, ALS, skeletal muscle, nutrition, exercise

## Abstract

With advancing aging, a decline in physical abilities occurs, leading to reduced mobility and loss of independence. Although many factors contribute to the physio-pathological effects of aging, an important event seems to be related to the compromised integrity of the neuromuscular system, which connects the brain and skeletal muscles via motoneurons and the neuromuscular junctions (NMJs). NMJs undergo severe functional, morphological, and molecular alterations during aging and ultimately degenerate. The effect of this decline is an inexorable decrease in skeletal muscle mass and strength, a condition generally known as sarcopenia. Moreover, several studies have highlighted how the age-related alteration of reactive oxygen species (ROS) homeostasis can contribute to changes in the neuromuscular junction morphology and stability, leading to the reduction in fiber number and innervation. Increasing evidence supports the involvement of epigenetic modifications in age-dependent alterations of the NMJ. In particular, DNA methylation, histone modifications, and miRNA-dependent gene expression represent the major epigenetic mechanisms that play a crucial role in NMJ remodeling. It is established that environmental and lifestyle factors, such as physical exercise and nutrition that are susceptible to change during aging, can modulate epigenetic phenomena and attenuate the age-related NMJs changes. This review aims to highlight the recent epigenetic findings related to the NMJ dysregulation during aging and the role of physical activity and nutrition as possible interventions to attenuate or delay the age-related decline in the neuromuscular system.

## 1. Introduction

The decline in physical ability and reduced mobility represent hallmarks of aging. While several events are known to contribute to muscle loss during aging, including chronic low-grade inflammatory status [1], reduced protein synthesis [2], mitochondrial dysfunction [3], and impaired satellite cell function [4], it is generally accepted that the decline in neural function is one of the principal factors leading to the pathological status known as sarcopenia [5,6,7]. Indeed, the nervous system not only controls muscle contraction and voluntary movements [8] but also affects myoblast orientation [9], skeletal muscle fiber specification, and myosin heavy chain (MHC) isoform expression [10]. Skeletal muscle fibers are classified as fast or slow according to the motor neuron activity, their distinct oxidative capacities, and ATPase activities [11]. These properties may be altered by a decrease in motor units, denervation of muscle fibers, and changes in NMJs [12]. Indeed, slow muscle phenotype is promoted by tonic motor neuron activity, while infrequent motor neuron firing results in fast fiber generation [12]. Moreover, cross reinnervation experiments demonstrated that fast muscles switch into slow ones when re-innervated by a slow nerve, whereas slow muscles switch into fast ones once re-innervated by fast nerve [13,14]. It has been reported by Klitgaard et al. that co-expression of different isoforms of MHC in a single muscle fiber increases with age, suggesting a continuous shift in the type of muscle fiber [15], and it is demonstrated that muscles in elderly individuals contain more slow muscle fibers that reduce contraction ability [16]. This also supports endplate fragmentation and functional denervation [17].

The inability of aged motor axons to efficiently re-innervate muscle fibers following bouts of degeneration [18] contributes to the atrophy of muscle fibers and loss of motor neurons. Motor neuron loss accompanied by muscle fiber loss may result in a consequent age-related increase in intermuscular fat and fibrous tissue [19,20], a minor degree of muscle fiber grouping [21], and a decrease in muscle size [6,22].

One possible mechanism of this progressive impairment of the re-innervation process seems related to age-associated degeneration of NMJ. NMJ is a synaptic connection where the peripheral nervous system contacts skeletal muscle fibers, thus controlling essential processes such as breathing and body voluntary movements [8]. Significant deterioration, including postsynaptic apparatus fragmentation, denervation, multiple innervations, and axon sprouting, has been observed in this critical region with advancing age [23,24,25]. Interestingly, similar defects characterize several severe motor neuron diseases such as amyotrophic lateral sclerosis (ALS), suggesting that strategies aimed to preserve NMJ integrity may be crucial in understanding the critical components involved in aging and ALS [26,27]. 

Although skeletal muscle fibers are characterized by having many nuclei, only those placed in the proximity of the NMJ are important for the transcription of the genes involved in NMJ structure, function, and maintenance [28,29,30]. The correct localization of synaptic nuclei at the NMJ requires the interplay between the cytoskeleton and several nuclear envelope proteins to effectively transduce cytoskeletal forces to the nucleus to regulate signaling pathways, chromatin organization, and gene expression [31,32,33,34]. It has been demonstrated that lamin A/C, an intermediate filament involved in establishing a continuous physical link between the nuclear lamina and the cytoskeleton [35,36,37,38], is reduced in aged muscles, and its mutations are implicated in premature aging disorders [39,40,41]. In particular, a recent paper published by Gao et al. demonstrated that muscle-specific lamin A/C mutation results in progressive denervation, AChR cluster fragmentation, and neuromuscular dysfunction [42]. These results demonstrate that lamin A/C is critical in the maintenance of NMJ and suggest that alterations in the cytoskeleton-nuclear network may contribute to NMJ degeneration in aged muscle.

Increasing evidence supports the involvement of epigenetic modifications in age-dependent morphological and functional alterations of the NMJ [25,43,44]. In particular, DNA methylation, histone modifications, and miRNA-dependent gene expression represent the major epigenetic mechanisms that play a crucial role in NMJ remodeling [45,46,47]. This review aims to highlight the recent epigenetic findings related to the NMJ dysregulation during aging and the role of physical activity and nutrition as possible interventions to attenuate or delay the age-related decline in the neuromuscular system. 

## 2. Age-Related Changes in NMJ 

Three principal elements compose NMJ: the presynaptic motor nerve terminals packed with synaptic vesicles containing the neurotransmitter acetylcholine (ACh); the synaptic cleft; and the deeply folded postsynaptic membrane of the muscle fiber, also referred to as the endplate, where acetylcholine receptors (AChRs) are located. When an action potential reaches the pre-synaptic element, voltage-dependent calcium channels open, allowing calcium to trigger the delivery of ACh in the synaptic cleft. Acetylcholine triggers AChR located in the post-synaptic membrane to produce an action potential, which in turn activates voltage-gated dihydropyridine receptors (DHPRs) located in the sarcolemma, and subsequently ryanodine receptors (RyRs) located in the sarcoplasmic reticulum membrane. The nerve terminal is covered by specialized glial cells, called terminal Schwann cells (tSCs), that produce a basal lamina that fuses with that of the muscle fiber at the boundary of the NMJ [48,49,50]. Moreover, fibroblast-like cells, known as kranocytes or perisynaptic fibroblasts, form a loose cover over the NMJ and participate in nerve repair and regeneration, thus representing other crucial players in neuromuscular functionality [50,51] (Figure 1). 

During aging, NMJs undergo dramatic morphological, functional, and molecular changes that lead them to ultimately degenerate [52,53,54]. At the level of the presynaptic region, increased axon diameter and larger nerve terminal area have been observed [23,55]. However, this effect is not accompanied by increased Ach stores, since it has been demonstrated that as presynaptic branching increases with age, the quantity of available Ach decreases [56,57]. At the postsynaptic level, the endplates decrease in size and are severely fragmented, and the number and length of postsynaptic folds are reduced, leading to a functional impairment of NMJ response [58]. Along with these modifications, a gradual decrease in the number of AChRs per junction has been observed [59]. Moreover, tSCs migrate away from the motor nerve terminal and protrude branches into the synaptic cleft, contributing to a functional decline of the neuromuscular system in aging [21,60]. 

A key role in the communication between nerve and muscle is played by the agrin-muscle-specific kinase (MuSk)–low-density lipoprotein receptor-related protein 4 (Lrp4) signaling pathway, which ensures the correct transmission of the action potential from the nerve terminal to the endplate, thus facilitating myofibrillar excitation [61,62]. It has been demonstrated that the agrin–MuSk–Lrp4 signaling pathway may be dysregulated with advancing aging. Agrin is expressed at NMJs by nerve terminals, muscle fibers, and tSC, where it is necessary for clustering of AchRs in muscle fibers [63,64,65,66,67,68]. In a mouse experimental model, Agrin impaired expression resulted in NMJ fragmentation, similar to what was observed in aged NMJs and premature sarcopenia [69]. Moreover, it has been demonstrated that the expression levels of NMJ-associated genes MuSk and Lrp4 appear to increase in aged muscle [70]. Interestingly, these genes are also upregulated in the denervated skeletal muscle of young mice, suggesting that aging may cause denervation of NMJs [71].

It has also been demonstrated that mitochondria dysfunction and oxidative stress, along with increased levels of intracellular calcium boost, determine the decline in NMJ [72,73], together with a reduced number of synaptic vesicles, similarly to what was observed in ALS [27,74,75,76]. 

## 3. ROS and NMJ Degeneration

Reactive oxygen species (ROS) are highly reactive molecules that are generated due to the electron receptivity of O_2_. In eukaryotic cells, endogenous production of ROS is associated with mitochondrial oxidative phosphorylation, by which a chain of redox reactions transports protons across the inner mitochondrial membrane, generating potential energy that accomplishes cellular ATP production [77].

One of the major molecules that play an important role in mitochondrial function is calcium (Ca2+). During muscle contraction, a greater influx of Ca2+ into the mitochondria occurs, which results in the activation of specific enzymes that lead to an increased synthesis of ATP to accomplish the energy demand. Moreover, increased mitochondrial Ca2+ levels have also been observed in several pathological conditions such as skeletal muscle denervation or unloading. The result of this steady-state elevation of mitochondrial Ca2+ level leads to increased ROS generation, induction of programmed cell death, and ultimately muscle atrophy [78].

In physiological conditions, a small number of electrons, passing through the chain, prematurely and incompletely reduces oxygen, producing the superoxide radicals; these are promptly scavenged by the cellular antioxidant enzymes, such as superoxide dismutase 1 (SOD1), to avoid oxidative damage of cellular macromolecules. Therefore, mitochondrial dysfunction and oxidative phosphorylation impairment determine the increase of ROS production and oxidative damage; these events, in turn, induce muscle cells to respond to the unbalanced redox homeostasis, upregulating the antioxidant enzymes [79]. Experimental evidence indicates that ROS may also act as chemical messengers, involved in receptor-mediated signaling pathways and transcriptional activation [80], suggesting that the cellular balance between ROS production and ROS scavenging is crucial to preserve cellular functionality [81].

Abnormal ROS levels contribute to the etiology of a wide variety of disorders, including aging and neurodegenerative diseases, such as ALS, and most of them are characterized by incoming defects in muscle–nerve communication [82].

Several animal models have been generated to elucidate the relation between oxidative damage and the alteration of the muscle–nerve endplate; among these are the transgenic mice overexpressing in skeletal muscle tissue the uncoupling protein 1 (UCP1), a mitochondrial protein that uncouples mitochondrial electron transport from ATP synthesis. These mice display deterioration of the NMJ, which correlates with progressive signs of muscle denervation and late-onset motor neuron pathology [83], demonstrating that a relatively mild mitochondrial dysfunction can recapitulate early signs of ALS disease, and it is sufficient to determine NMJ alteration and motor neuron degeneration.

Several other studies have investigated the relationship between alteration of antioxidant defense and NMJ defects. Data from SOD1 knockout mice (SOD1^−/−^) [84] and mutant SOD1 gain-of-function models (SOD1^G93A^ mice) [85,86] demonstrate the importance of SOD1 enzyme activity and ROS homeostasis in muscle–nerve communication. Indeed, it has been shown that the ubiquitous expression of the mutant human SOD1^G93A^ gene induces a widespread spinal motor neuron death in mice. Conversely, the genetic ablation of the SOD1 gene (SOD1^−/−^) in mouse determines, at advanced ages, a motor neuropathy associated with distal motor axon degeneration, without loss of ventral motor neurons, confirming the crucial role of the SOD1 antioxidant activity in the maintenance of NMJ stability. Interestingly, muscle denervation in SOD1^−/−^ strongly correlates with loss of mitochondria at the motor nerve terminal during postnatal life. The defect is rescued by the replacement of SOD1 enzyme in the mitochondrial inner membrane [87], indicating the close relationship between the alteration of mitochondria functionality, induced by oxidative stress, and defects in muscle–nerve communication.

Although it is widely accepted that NMJ alteration during aging or ALS disease is closely related to axon and synapse damage, controversy exists on whether pathological events could begin at the skeletal muscle, potentially influencing the loss of NMJs and the degeneration of motor neurons. Indeed, the temporal analysis of axon and neuromuscular junction degeneration in an ALS mouse model indicated that motor neuron degeneration starts distally at the NMJ area and occurs earlier than clinical symptoms, proceeding towards neuron soma in a retrograde dying back manner [88]. 

The transgenic mice MLC/SOD1G93A, overexpressing the SOD1G93A gene under the control of a muscle-specific promoter, provide further evidence to support the dying-back theory and the concept that skeletal muscle is a primary target of the toxic effect of the SOD1 mutant gene. Indeed, the transgenic MLC/SOD1G93A mice exhibit defects of the mitochondrial membrane potential and reduced integrity of the mitochondrial network, which are associated with higher turnover and fragmentation of the NMJ [75]. This evidence is corroborated by a recent work of Wong and colleagues who demonstrated that skeletal muscle-restricted expression of wild-type or mutant hSOD1 genes causes age-related ALS-like phenotype [89], that is, motor defects, distal axonopathy, NMJ pathology, and motor neuron loss, according to a non-autonomous mechanism for motor-neuron degeneration [90]. 

The important role of the antioxidant enzymes in muscle and nerve communication has been investigated in several studies focused on ROS production during muscle contraction [91]. Contractile activity in muscles of young rodents leads to the extracellular release of ATP and the activation of an NADPH oxidase (NOX2) complex in the muscle post-synaptic terminal. NOX2 is responsible for superoxide and hydrogen peroxide (H_2_O_2_) production and regulates, together with ATP, the release of ACh from the motor nerve terminal [92]. NOX2 activates redox-sensitive transcription factors that in turn induce higher expression of antioxidant enzymes, such as SOD and catalase, to counteract oxidative damage and facilitate exercise-induced muscle tissue remodeling [93]. It has been reported that exercise, through the activation of mitogen-activated protein kinases (MAPKs) and through nuclear factor kappa-light-chain-enhancer of activated B cell (NF-κB) signaling pathways, can cause the expression of antioxidant enzymes that play a crucial role in the protection against ROS, as well as adaptation to exercise [94,95]. However, the beneficial effects of physical activity are lost with exhaustive endurance and resistance exercise since the increased levels of ROS observed in these conditions overwhelm cellular antioxidant defenses, leading to tissue damage [96,97,98,99]. Excessive ROS production during aging or neurodegenerative disorder is responsible not only for mitochondria damage but also for cellular protein oxidation (Figure 2). Elderly muscles accumulate damaged proteins and organelles, since the activity of the autophagy system, a regulated mechanism for the removal of unnecessary or dysfunctional components, declines during aging. Indeed, in a recent paper [100], the authors described how the inhibition of the autophagic flux enhances oxidative stress in young muscles and alters muscle force generation, inducing mitochondrial dysfunction and NMJ instability; in this context, autophagy emerges as a required mechanism for mitochondrial quality control and the correct interplay between muscle and nerve [101]. 

In conclusion, this evidence demonstrates that the age-related alteration of ROS homeostasis affects muscle–nerve communication; therefore, the maintenance of redox balance can be considered as a good therapeutic tool to counteract the decline of NMJ functionality of elderly individuals.

## 4. Epigenetic Regulation of NMJ Dysfunction

In recent years, many works have shown that epigenetic changes can be responsible for the development of age-related diseases [43,102,103]. Among the major epigenetic mechanisms, DNA methylation, histone acetylation, and miRNA-dependent gene expression have an important role in the regulation of signals between motoneurons and NMJ.

### 4.1. DNA Methylation

One of the most extensively investigated epigenetic mechanisms in aging is DNA methylation, which results in the covalent addition of a methyl group to the cytosine in a CpG dinucleotide. The methylation of CpG-rich regions placed nearby the promoter region of a specific gene results in gene silencing [43]. On contrary, DNA methylation of the gene body is found in actively transcribed genes [103,104,105]. Several findings support CpG methylation deregulation as contributing factor to age-related decline in synaptic plasticity and homeostasis [106,107]. Three catalytically active enzymes, DNA methyltransferases (DNMT) DNMT1, DNMT3a, and DNMT3b, have been identified for the establishment and maintenance of DNA methylation patterns in mammalian cells [108]. 

It has been demonstrated that the cytokine TNF-like weak inducer of apoptosis 2 (TWEAK2) and its receptor Fn14 are important mediators of skeletal muscle atrophy that occurs under denervation conditions [109,110]. Elevated activity of TWEAK2–Fn14 system in denervated muscle results in the activation of the ubiquitin–proteasome system, leading to muscle atrophy [109]. Tajrishi et al. demonstrated that in denervated skeletal muscle, Fn14 gene expression is regulated through DNA methylation. Denervation of skeletal muscle leads to reduced expression of DNMT3a and hypo-methylation of multiple CpG sites in Fn14 promoter, while overexpression of DNMT3a inhibits the expression of Fn14 and reduces atrophy in denervated skeletal muscle [108].

Although the molecular mechanism by which methylation modulates aging is still unknown, the results obtained from different experimental models such as Drosophila and mammals highlight a correlation between DNA methylation and aging and suggest that DNA methylation could be used as a general biomarker for aging [111].

### 4.2. Histone Acetylation

Histone acetylation is generally associated with transcriptional activation of chromatin and is regulated by the opposite activities of histone acetyltransferases (HATs) and histone deacetylases (HDACs). In particular, HATs catalyze the transfer of acetyl groups to lysine residues of histones, resulting in the relaxation of chromatin, allowing the transcription. On the other hand, HDACs can reverse histone acetylation, leading to chromatin condensing, finally preventing the transcriptional activation of muscle-specific genes [112]. A number of studies have highlighted the crucial role of HAT and HDAC in the regulation of skeletal muscle development and differentiation. In particular, experimental evidence revealed that HDAC4 expression increases following denervation of muscular fibers and negatively correlates with NMJ function [113]. Furthermore, high HDAC4 expression levels were detected in a mouse model of spinal muscular atrophy in ALS patients and in an ALS mouse model, suggesting that in the presence of neuromuscular defects, inactivation of HDAC4 may result in a suppression of muscle atrophy in a denervation model [113,114]. 

Since several studies performed in both rodent model of ALS (G86R mutant SOD1 mice) [115] and human postmortem samples have demonstrated a downregulation of histone acetylation in ALS, it has been hypothesized that the maintenance of proper acetylation by using HDAC inhibitors would be beneficial in ALS and in other neurodegenerative diseases [31,116,117]. For example, it has been reported that the HDAC inhibitor trichostatin A induced a modest improvement in motor function and survival as well as protection against motor neuron death, axonal degeneration, muscle atrophy, and neuromuscular junction denervation in an ALS mouse model [118]. Another HDAC inhibitor is the valproic acid (VPA) that, when administered to transgenic ALS SOD1 mice, has been shown to not improve survival or motor performance but to slightly delay the onset of motor decline, without being able to prevent disruption of NMJ [119,120]. However, in an ALS human patient, VPA treatment did not show beneficial effects on disease progression or survival prolongation [119]. 

Phenyl butyrate (PBA), another HDAC inhibitor, resulted in significant improvement in clinical and neuropathological phenotypes of the SOD1 mice. PBA restored histone acetylation levels, increased motoneuron survival, life extent, and motor performances, as compared with vehicle-treated animals, and, importantly, was effective even when administered at disease onset [121,122].

Sirt1 is a nicotinamide adenine dinucleotide (NAD+)-dependent deacetylase, involved in ALS and in other neurodegenerative diseases, where it plays a crucial role in attenuating damages caused by aging [123,124]. Overexpression of Sirt1 in the brain of aged mice correlates with a higher percentage of innervated NMJs, less fragmented AChR clusters, and an increased number of NMJs covered by tSCs [125,126]. Indeed, in a recent paper by Herskovits et al., it was demonstrated that Sirt1 expression decreased in the spinal cord of wild-type mice during normal aging, and the overexpression of Sirt1 in motor neurons partially protected NMJs from the deleterious effects of aging and ALS [125].

### 4.3. miRNAs

miRNAs are ≈22-nucleotide small RNAs that bind to complementary regions on numerous mRNAs, thereby regulating their expression [127,128,129]. MiRNAs are required for many biological processes, such as intercellular communication, differentiation, and proliferation [130,131]. Several miRNAs, including miRNA-1, miRNA-206, miRNA-208b, miRNA-133a, miRNA-133b, miRNA-486, and miRNA-499, known as myomiRNAs, are specifically expressed in skeletal muscle [132,133,134], where they regulate many physiological processes, such as growth, development, and maintenance of muscle mass and function [135,136,137]. Consequently, alterations of miRNA expression may occur during aging and can be associated with pathological conditions [136,138,139,140,141]. 

It has been demonstrated by Valdez et al. that miR-206, known to play an important role in muscle recovery and regeneration, is also crucial in the formation of new NMJs following nerve injury [23]. In particular, miR-206 expression levels are upregulated following denervation and in mouse models of muscular dystrophy and ALS. Moreover, in the absence of miR-206, the reinnervation process is slowed down following nerve injury, and the progression of the disease is accelerated in the mouse model of ALS [114,142]. These data highlighted a crucial role for miR-206 to promote the reinnervation process and to counteract the progression of the disease. Moreover, miR-206 has been found to repress the HDAC4 expression in muscle and to induce the expression of the secreted factor fibroblast growth factor binding protein 1 [114], thus counteracting the negative effect of HDAC4 on reinnervation [143].

miR-133b, another myomiRNA known to be involved in skeletal muscle development and function, has also been shown to be enriched at the NMJ, and its expression increases following denervation [135]. miR-133b has been proposed to have a role in the NMJ maintenance; however, as reported by Valdez et al., an miR-133b null mouse showed no difference in NMJ morphology and the reinnervation of the muscle fibers [144]. In a very recent paper by Yang et al., the authors demonstrated that in a mouse sciatic nerve transection model, miRNA-142a-5p functioned as an important regulator of denervation-induced skeletal muscle atrophy by inducing mitochondrial dysfunction, mitophagy, and apoptosis by targeting membrane-anchored protein mitofusin-1 (MFN1) [145]. Other miRNAs that have been demonstrated to have an important function in the regulation of the quality and quantity of NMJs and motoneurons are miR-146a and miR-234 [140]. In particular, in a mouse model of spinal muscular atrophy, miR-146a overexpression results in loss of motoneurons, while its abrogation prevents motoneurons loss [146]. Further analyses revealed that mir-234 may be responsible for NMJ dysfunctions through its ability to control the expression of target genes that are important for the maturation and transport of neuropeptides in *Caenorhabditis elegans* [147]. Although more research is needed, these studies highlight a possible role for miRNAs in regulating neurodegeneration in sarcopenia (Table 1).

### 4.4. Nutrition-Dependent Epigenetic Regulation of NMJ 

Nutrition is one of the most important environmental factors involved in the modulation of epigenetic events. Many nutrients and diet compounds have been demonstrated to affect DNA methylation, histone acetylation/deacetylation status, and miRNAs expression by controlling the activity of epigenetic modifying enzymes [148]. Thus, it is possible to hypothesize a role for nutrition to counteract or eventually reverse the aberrant epigenetic phenomena related to age-associated diseases (Figure 3). 

In recent years, several studies have highlighted a potential neuroprotective role of satellite cells during aging. Satellite cells represent the adult muscle resident stem cells whose activity is indispensable for muscle regeneration [149,150,151], although during aging their number and proliferative activity dramatically decrease [152,153,154]. It has been shown by Liu et al. that satellite cell deficiency worsens muscle atrophy and aggravates deficits in reinnervation and post-synaptic morphology at regenerating NMJs [155,156]. Moreover, a dramatic decrease in satellite cell content was also found after 14 days of bed rest with concomitant significant skeletal muscle atrophy [157]. These data highlight that a long period of inactivity significantly affects satellite cell function and suggests that controlling and increasing satellite cell number during aging may represent an efficient strategy to preserve NMJ health, delaying the loss of muscle mass and function. These severe alterations of satellite cell function, which occur with advancing aging, can be the result of either extrinsic or intrinsic events such as impaired self-renewal mechanisms, exhaustion following forced differentiation, and alteration of muscle environment [158,159,160]. In 2005, heterochronic transplantation experiments performed in the Conboy laboratory demonstrated that aged satellite cells exposed to a young systemic environment showed dramatic improvements in functionality [161]. Moreover, it has been shown that specific nutrients may also enhance satellite cell activity by promoting a more rejuvenating systemic milieu both in in vivo and in vitro experimental models [162]. These data highlight a crucial role of the environmental milieu including dysregulated signals from either myofiber or the circulatory system in the homeostasis of skeletal muscle tissue. It has been well established that the consumption of specific nutrients such as essential amino acids (EAAs) and glucose may alter the expression of miRNAs involved in satellite cell function [163,164,165,166,167]. Indeed, it has been demonstrated by Zhang et al. that nutrient-dependent stimulation of mTOR signaling may affect the expression of miR-133a/b and miR-206 that, as previously mentioned, has been proposed to have a crucial role in the satellite cell and NMJ maintenance [141,168].

There are a number of natural compounds that have been demonstrated to have antioxidants activity [169,170,171,172,173,174,175]. Vitamin D is a fat-soluble vitamin present in many foods, although it is primarily produced in the skin when it has been exposed to ultraviolet (UV) rays [176]. The reduction of muscle mass associated with advancing aging has been clearly associated with decreased circulating vitamin D levels, leading to frailty in the elderly and frequent falls [177,178,179,180,181]. It was reported that in a mouse model, vitamin D deficiency led to alteration in NMJ-related genes and protein expression levels [182]. Vitamin E, of which many vegetables such as tomatoes, broccoli, and spinach are rich in, has been demonstrated to prevent lipid peroxidation and to counteract the negative effects of free radicals in cellular membranes and lipoproteins, revealing protective functions against neurodegenerative diseases [183].

In mice fed with sodium butyrate, a natural bacterial product, intestinal microbial homeostasis was restored, gut integrity was improved, and life span was prolonged compared with those of control mice. In both ALS mice and intestinal epithelial cell cultures derived from humans, sodium butyrate treatment was associated with decreased aggregation of the SOD1 mutated protein. The findings from this study highlight the complex role of the gut microbiome and intestinal epithelium in the progression of ALS and indicate sodium butyrate as a potential therapeutic reagent for restoring ALS-related dysbiosis [184].

In addition, the histone deacetylase inhibitor PBA significantly improved motor function and extended survival in the G93A transgenic mouse model. This effect was enhanced when PBA was administrated in combination with the antioxidant AEOL 10150, suggesting that HDAC inhibitors and blocking oxidative stress agents may exert additive therapeutic effects in treating mutant SOD1-associated ALS and other neurodegenerative diseases [121].

Another interesting study demonstrated that caloric restriction and resveratrol, a small natural polyphenol, which is a caloric restriction mimetic, can attenuate NMJ fragmentation and degradation in a mouse model of ALS [23,185]. In particular, it has been shown that resveratrol, when used at certain doses, is capable of slowing NMJ degeneration, while at other doses it does not affect the disease progression [185,186,187,188,189,190,191].

### 4.5. Exercise-Dependent Regulation of NMJs

It is well known that the decreased physical activity that occurs with advancing aging may contribute to the onset of sarcopenia [192]. Indeed, several studies demonstrate that exercise has beneficial effects on muscle homeostasis and force production by the modulation of signaling pathways involved in fiber-type specification, muscle growth, and NMJ remodeling [193,194,195]. In particular, several studies performed in experimental mouse models revealed that physical exercise improves age-dependent NMJ function by decreasing their fragmentation and denervation [23,193]. The importance of physical exercise in NMJ maintenance is further supported by the evidence that during periods of chronic inactivities, such as bed rest and space flight, increased serum levels of NMJ instability and deterioration markers have been detected [196,197,198]. However, how exercise may exert its beneficial effect in counteracting age-dependent NMJ dismantlement is still not fully elucidated.

Under voltage-clamp condition, it has been found that the loss of physiological Ca2+ transients or mitochondrial Ca2+ uptake could be an initial trigger for mitochondrial dysfunction with increased mitochondrial ROS production in skeletal muscle fibers following denervation [199]. Both acute and long-term endurance exercises have been reported to activate certain signaling pathways to counteract ROS production. Meanwhile, electrical stimulation is known to help prevent apoptosis and alleviate muscle atrophy in denervated animal models and patients with motor impairment. Several studies focus on the excitation–transcription coupling framework to understand the beneficial role of exercise and electrical stimulation. Interestingly, a recent study has revealed an unexpected role of rapid mitochondrial Ca2+ transients in keeping mitochondrial permeability transition pore (mPTP) at a closed state with reduced mitochondrial ROS production [199].

We must consider that skeletal muscle has been described as an organ able to produce and secrete specific molecules, known as myokines, that exert autocrine, paracrine, and endocrine regulatory functions on different organs and tissues including the nervous system [200,201,202].

As previously mentioned, ROS production during physical activity modules signaling pathways involved in muscle remodeling and in the adaptive response necessary to control oxidative stress [97]. Among the principal redox-sensitive signaling pathways that play a very important role in the skeletal muscle adaptive response to oxidative stress are NF-κB, MAPKs, and PGC1α [203]. In particular, NF-kB induced by increased levels of hydrogen peroxide and/or inflammatory cytokines leads to high levels of molecules such as iNOS, cyclooxygenase 2 (COX2), and SOD2 [204], while ROS-dependent activation of MAPK signaling results in increased glucose uptake by muscle fibers during exercise [205]. Another important factor involved in relaying the positive effect of exercise on skeletal muscle and NMJ is PGC1α. Indeed, in old transgenic mice, the overexpression of PGC1α results in increased levels of genes involved in NMJ function and morphology and induces the shift from fast to slow phenotype of muscle fiber [206,207,208]. Moreover, the abrogation of PCG1α expression in skeletal muscle inhibits the expression of several NMJ genes and the clustering of AChRs [207]. Among other factors that play a positive role in counteracting the effect of aging on muscle and NMJs, there is insulin-like growth factor-1 (IGF-1). Indeed, in senescent mice, muscle-specific overexpression of IGF-1 results in the activation of molecular signaling pathways involved in the maintenance of NMJ integrity [209,210,211,212].

## 5. Conclusions and Future Perspective

This review highlights the recent epigenetic findings related to the NMJ dysregulation during aging and/or neuromuscular disorders. In addition, it discusses the role of physical activity and nutritional intervention as potential tools to ameliorate aging-associated decline or degenerative disease conditions by protecting and maintaining NMJ integrity. Future studies are needed to better clarify the molecular mechanisms underpinning lifestyle influence on NMJs homeostasis in order to develop new therapeutic strategies to counteract the negative effect of aging in the neuromuscular system.

## Figures and Tables

**Figure 1 cells-10-01307-f001:**
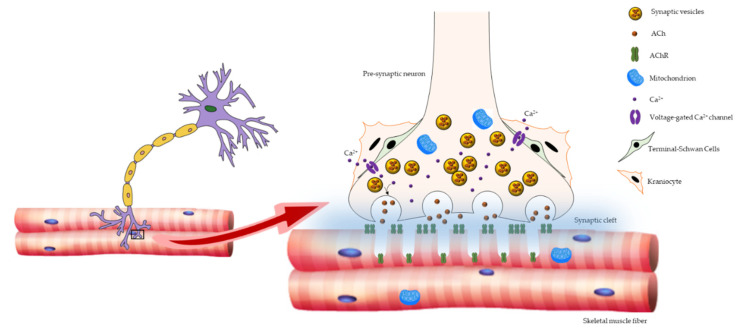
Schematic representation of NMJ architecture. NMJ is composed of three major elements: the presynaptic motor nerve terminals packed with synaptic vesicles containing the neurotransmitter ACh; the synaptic cleft; and the deeply folded postsynaptic membrane of the muscle fiber, where AChRs are placed. The NMJ-associated cells, tSCs, and kranocytes are also shown.

**Figure 2 cells-10-01307-f002:**
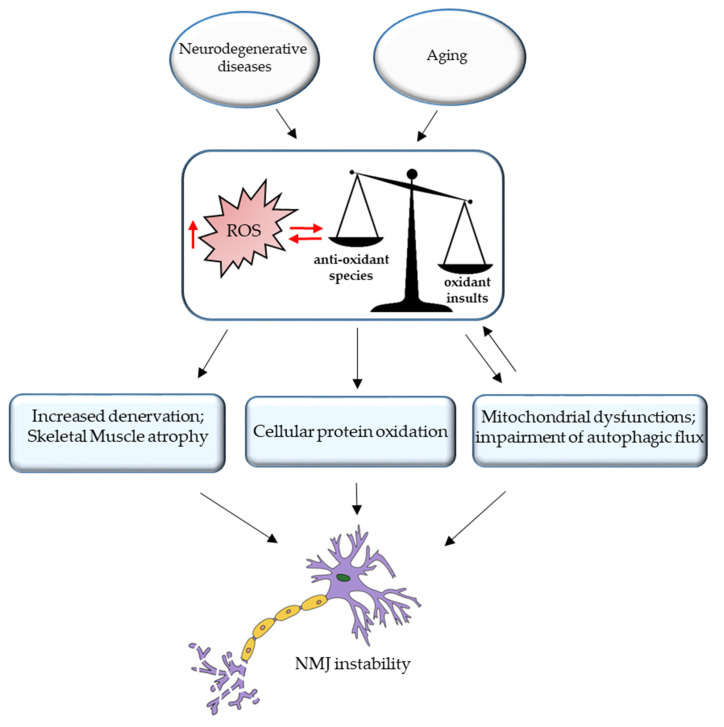
Oxidative stress and NMJ instability. Excessive ROS production during aging or neurodegenerative disorder is responsible for the increased denervation and skeletal muscle atrophy, cellular protein oxidation, mitochondrial dysfunction, and impairment of autophagic flux, leading to NMJ instability.

**Figure 3 cells-10-01307-f003:**
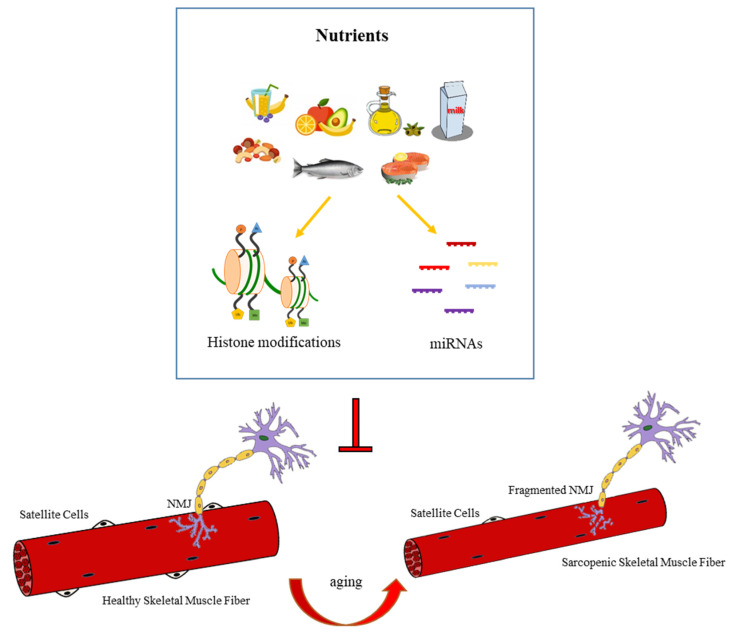
Nutrient-dependent epigenetic modifications involved in NMJ remodeling during aging. Nutrients and diet compounds may affect histone modifications and miRNAs expression, counteracting NMJ dismantlement involved in the onset of sarcopenia.

**Table 1 cells-10-01307-t001:** Table showing some of the miRNAs involved in aging and/or neurodegenerative diseases.

miRNAs	Experimental Models	Role	References
miR-206	Following nerve injury and in mouse models of muscular dystrophy and ALS.	Promotes the reinnervation process and counteracts the progression of the diseases.	[113,141,142]
miR-133b	Sciatic nerve excision and mouse model of ALS.	It is not required to maintain or restore NMJs following acute nerve injury or in a motor neuron disease.	[143]
miRNA-142a-5p	Sciatic nerve excision	It is an important regulator of denervation-induced skeletal muscle atrophy.	[144]
miR-146a	In a mouse model of spinal muscular atrophy.	Its overexpression results in loss of motoneurons while its abrogation prevents motoneurons loss.	[139,145]
miR-234	In *Caenorhabditis elegans*	Its overexpression endows resistance to the Ach inhibitor suggesting modification of NMJ function.	[146]

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
