# Peer review of "Age-Related Alterations at Neuromuscular Junction: Role of Oxidative Stress and Epigenetic Modifications"

_cells, 2021, doi:10.3390/cells10061307_

Round 1

Reviewer 1 Report

This review by Dobrowolny et al is gives a fairly comprehensive and well-written overview of the contributions of epigenetic factors involved in NMJ decline which result in sarcopenia. The one critique I have is that the authors do not mention the contributions of aging nuclear envelope and lamin A/C architecture can have on the NMJ. I would suggest a brief overview of this subject. 

Author Response

We are very grateful to the Reviewers for giving us the opportunity to improve the quality of our work. Their suggestions were taken into careful consideration. We feel we were able to adequately address the issues raised by the comments.

Reviewer 1

Minor comments:

  • This review by Dobrowolny et al gives a fairly comprehensive and well-written overview of the contributions of epigenetic factors involved in NMJ decline which result in sarcopenia. The one critique I have is that the authors do not mention the contributions of the aging nuclear envelope and lamin A/C architecture can have on the NMJ. I would suggest a brief overview of this subject.

As suggested by Reviewer 1, we introduced a brief overview of the role of the nuclear envelope (NE) and lamin A/C on NMJs structure, and function during aging. In particular, we added in the Introduction section, page. 2, lines 68-81:

“Although skeletal muscle fibers are characterized by having many nuclei, only those placed in the proximity of the NMJ are important for the transcription of the genes involved in NMJ structure, function, and maintenance [1–3]. The correct localization of synaptic nuclei at the NMJ requires the interplay between the cytoskeleton and several nuclear envelope proteins to effectively transduce cytoskeletal forces to the nucleus regulating signaling pathways, chromatin organization, and gene expression [4–7]. It has been demonstrated that lamin A/C, an intermediate filament involved in establishing a continuous physical link between the nuclear lamina and the cytoskeleton [8–11],  is reduced in aged muscles and its mutations are implicated in premature aging disorders [12–14]. In particular, in a  recent paper published by Gao et al., it was demonstrated that muscle-specific lamin A/C mutation results in progressive denervation, AChR cluster fragmentation, and neuromuscular dysfunction [15]. These results demonstrate that lamin A/C is critical in the maintenance of NMJ and suggest that alterations in the cytoskeleton-nuclear network may contribute to NMJ degeneration in aged muscle.

References

  1. Li, L.; Xiong, W.C.; Mei, L. Neuromuscular Junction Formation, Aging, and Disorders. Annu. Rev. Physiol. 2018, 80, 159–188.
  2. Ruegg, M.A. Organization of synaptic myonuclei by Syne proteins and their role during the formation of the nerve-muscle synapse. Proc. Natl. Acad. Sci. U. S. A. 2005, 102, 5643–5644.
  3. Schaeffer, L.; De Kerchove D’Exaerde, A.; Changeux, J.P. Targeting transcription to the neuromuscular synapse. Neuron 2001, 31, 15–22.
  4. Battey, E.; Stroud, M.J.; Ochala, J. Using nuclear envelope mutations to explore age-related skeletal muscle weakness. Clin. Sci. 2020, 134, 2177–2187.
  5. Grady, R.M.; Starr, D.A.; Ackerman, G.L.; Sanes, J.R.; Han, M. Syne proteins anchor muscle nuclei at the neuromuscular junction. Proc. Natl. Acad. Sci. U. S. A. 2005, 102, 4359–4364.
  6. Saha, R.N.; Pahan, K. HATs, and HDACs in neurodegeneration: A tale of disconcerted acetylation homeostasis. Cell Death Differ. 2006, 13, 539–550.
  7. Zhang, X.; Xu, R.; Zhu, B.; Yang, X.; Ding, X.; Duan, S.; Xu, T.; Zhuang, Y.; Han, M. Syne-1 and Syne-2 play crucial roles in myonuclear anchorage and motor neuron innervation. Development 2007, 134, 901–908.
  8. Hutchison, C.J. Lamins: Building blocks or regulators of gene expression? Nat. Rev. Mol. Cell Biol. 2002, 3, 848–858.
  9. Burke, B.; Stewart, C.L. The nuclear lamins: Flexibility in function. Nat. Rev. Mol. Cell Biol. 2013, 14, 13–24.
  10. Burke, B.; Stewart, C.L. The laminopathies: The functional architecture of the nucleus and its contribution to disease. Annu. Rev. Genomics Hum. Genet. 2006, 7, 369–405.
  11. Mounkes, L.; Kozlov, S.; Burke, B.; Stewart, C.L. The laminopathies: Nuclear structure meets disease. Curr. Opin. Genet. Dev. 2003, 13, 223–230.
  12. De Sandre-Giovannoli, A.; Chaouch, M.; Kozlov, S.; Vallat, J.M.; Tazir, M.; Kassouri, N.; Szepetowski, P.; Hammadouche, T.; Vandenberghe, A.; Stewart, C.L.; et al. Homozygous defects in LMNA, encoding lamin A/C nuclear-envelope proteins, cause autosomal recessive axonal neuropathy in human (Charcot-Marie-Tooth disorder type 2) and mouse. Am. J. Hum. Genet. 2002, 70, 726–736.
  13. Broers, J.L.V.; Ramaekers, F.C.S.; Bonne, G.; Ben Yaou, R.; Hutchison, C.J. Nuclear lamins: Laminopathies and their role in premature ageing. Physiol. Rev. 2006, 86, 967–1008.
  14. Scaffidi, P.; Misteli, T. Lamin A-dependent nuclear defects in human aging. Science (80-. ). 2006, 312, 1059–1063.
  15. Gao, N.; Zhao, K.; Cao, Y.; Ren, X.; Jing, H.; Xing, G.; Xiong, W.C.; Mei, L. A role of lamin A/C in preventing neuromuscular junction decline in mice. J. Neurosci. 2020, 40, 7203–7215.

Reviewer 2 Report

Review comments on Manuscript ID: cells-1218793

This review article by Dobrowolny et al covers the “Role of Nutrition and Exercise to counteract Granbury, TexasSarcopenia and Age-Related Alterations at Neuromuscular Junction” with aims to highlight the recent epigenetic findings related to the NMJ dysregulation during aging and the role of physical activity and nutrition as possible interventions to attenuate or delay the age-related decline in the neuromuscular system. It is an attractive topic that covers exciting and expanding frontiers in the field of understanding neuromuscular degeneration, although there are some concerns on the depth of the discussion, the accuracy of the reference citation, and potential overstated conclusions.

  1. In the discussion concerning effect of mitochondrial dysfunction in determining the decline in NMJ, please include original studies: (doi: 10.1074/jbc.M109.041319 and DOI: 10.1152/ajpregu.00767.2006).

  1. In the section concerning ROS and NMJ degeneration, please at least briefly discuss the link between mitochondrial Ca2+ signaling and ROS production (doi: 10.3389/fphys.2020.595800).

  1. In the section concerning DNA methylation, please at least briefly discuss about functional or expressional changes of DNA methyl-transferase (DNMT) linked to aging or neuromuscular degenerative disease, such as DNMT3a (doi: 10.1074/jbc.M114.568626).

  1. In the section concerning HDAC, the authors may further elaborate on the potential application of HDAC inhibitors (HDACi) in the treatment of neuromuscular degenerative disease. For example, the de-repression of myogenic genes and miRNAs in fibroadipogenic progenitors (doi: 10.1101/gad.234468.113).

  1. In the section concerning miRNAs, although miRNAs predominantly work through the mechanism described by the authors, there do exist exceptions in which miRNA positively regulate the translation of the target mRNA, such as miR-10a (doi: 10.1016/j.molcel.2008.05.001). Thus, the first sentence needs to be rephrased. Furthermore, beside miR-206, miR-133b, miR-146a, please also discuss the recent work concerning miR-142a-5p/MFN1 axis (doi: 10.7150/thno.40857).

  1. In the section concerning nutrition, it is worthwhile to mention certain diet supplement, such as sodium butyrate and phenylbutyrate (both are HDAC inhibitors), have been associated with the alleviation of ALS in animal models (doi: 10.1016/j.clinthera.2016.12.014; doi: 10.1016/j.nbd.2005.09.013)

  1. In the section concerning exercise, the impact of physiological Ca2+ transients on muscle ROS production could be included (doi: 10.1186/s13395-017-0123-0).

  1. Page 2, line 67-69: “…DNA methylation, histone modification, and miRNAs-dependent gene expression represent the major epigenetic mechanisms that play a crucial role in NMJs remodeling [19-21]”. Those three cited references have no discussion on NMJ remodeling.

  1. Page 6, line 244-245: "…overexpression of Sirt1 in the brain of aged mice correlates with a higher percentage of innervated NMJs, less fragmented AChR clusters, and in increased number of NMJs…[74,75]”. Again, those two cited references (original studies) have no data on NMJ.

  1. Page 6, line 259: a reference number should be included for Valdez G. et al

  1. Page 6: Line 259-261: the citation 91 and 92 have no study on the formation of new NMJs following nerve injury.

  1. Page 7: Line 317-320: “Another interesting study demonstrates that caloric restriction and resveratrol, which is a caloric restriction mimetic, can attenuate NMJ fragmentation and degradation partially through the Sirt1, peroxisome proliferator-activated receptor-gamma coactivator alpha (PGC1α), and mTOR signaling pathways [119–122].”  None of those listed citations has studies on NMJ.  Additionally, when discussing the beneficial effect of resveratrol, the authors should also be aware and discuss the controversial experimental evidence on resveratrol, to avoid a potential overstatement and misleading readers. 

Minor issues:

Please add in the legend of Fig. 2 that CAT is the abbreviation of catalase.

In the section concerning exercise, PGC1α was misspelled as PCG1α twice.

Author Response

Major comments:

  • In the discussion concerning effect of mitochondrial dysfunction in determining the decline in NMJ, please include original studies: (doi:10.1074/jbc.M109.041319 and DOI: 10.1152/ajpregu.00767.2006).

As suggested by Reviewer 2, we added the indicated original studies in “Age-related changes in NMJ” section, page 3, line 131 (Ref. 73-74 of the manuscript).  

  • In the section concerning ROS and NMJ degeneration, please at least briefly discuss the link between mitochondrial Ca2+ signaling and ROS production (doi: 10.3389/fphys.2020.595800).

As suggested by Reviewer 2, in the new version of the manuscript, we added the link between mitochondrial Ca2+ signaling and ROS production with the indicated reference. In particular, page 4, lines 144-151 (Ref. 79 of the manuscript):

One of the major molecules that play an important role in mitochondrial function is calcium (Ca2+). During muscle contraction, a greater influx of Ca2+ into the mitochondria occurs which results in the activation of specific enzymes that lead to an increased synthesis of ATP to accomplish the energy demand. Besides, increased mitochondrial Ca2+ levels have been also observed in several pathological conditions such as skeletal muscle denervation or unloading. The result of this steady-state elevation of mitochondrial Ca2+ level leads to increased ROS generation, induction of programmed cell death, and ultimately muscle atrophy [1].

  • In the section concerning DNA methylation, please at least briefly discuss about functional or expressional changes of DNA methyl-transferase (DNMT) linked to aging or neuromuscular degenerative disease, such as DNMT3a (doi: 10.1074/jbc.M114.568626).

As suggested by Reviewer 2, in the new version of the manuscript, we discussed this topic and introduced the new citation. In particular, pages 6 and 7, lines 250-261 (Ref.107 of the manuscript):

“Three catalytically active enzymes, DNA methyltransferases (DNMT), DNMT1, DNMT3a, and DNMT3b, have been identified for the establishment and maintenance of DNA methylation patterns in mammalian cells [2]. It has been demonstrated that the cytokine TNF-like weak inducer of apoptosis 2 (TWEAK2) and its receptor Fn14 are important mediators of skeletal muscle atrophy that occurs under denervation conditions [3,4]. Elevated activity of TWEAK2-Fn14 system in denervated muscle results in the activation of the ubiquitin-proteasome system leading to muscle atrophy [3]. Tajrishi et al. demonstrated that in denervated skeletal muscle, Fn14 gene expression is regulated through DNA methylation. Denervation of skeletal muscle leads to reduced expression of DNMT3a and hypomethylation of multiple CpG sites in Fn14 promoter while overexpression of DNMT3a inhibits the expression of Fn14 and reduces atrophy in denervated skeletal muscle [2]”.

  • In the section concerning HDAC, the authors may further elaborate on the potential application of HDAC inhibitors (HDACi) in the treatment of neuromuscular degenerative disease. For example, the de-repression of myogenic genes and miRNAs in fibroadipogenic progenitors (doi: 10.1101/gad.234468.113).

As suggested by the Reviewer, in the new version of the manuscript, we discuss this topic. However, the Authors believe that the suggested reference does not appear to be strictly correlated with the role of HDACi in NMJs remodeling. Therefore, we introduced the following with the appropriate references. In particular, page 7, lines 283-300 (Ref. 32, 114-121 of the manuscript):

“Since several studies performed in both rodent model of ALS (G86R mutant SOD1 mice) [5] and human postmortem samples have demonstrated downregulation of histone acetylation in ALS, it has been hypothesized that the maintenance of proper acetylation by using HDAC inhibitors would be beneficial in ALS and in other neurodegenerative diseases [6–8]. For example, it has been reported that the HDAC inhibitor Trichostatin A induced a modest improvement in motor function and survival as well as protection against motor neuron death, axonal degeneration, muscle atrophy, and neuromuscular junction denervation in ALS mouse model [9]. Another HDAC inhibitor is the valproic acid (VPA) that, when administered to transgenic ALS SOD1 mice, has been shown to not improve survival or motor performance but to slightly delay the onset of motor decline, without being able to prevent disruption of NMJ [10,11]. However,  in an ALS human patient VPA treatment did not show beneficial effects on disease progression or survival prolongation [10]. Phenyl Butyrate (PBA), another HDAC inhibitor, resulted in significant improvement in clinical and neuropathological phenotypes of the SOD1 mice. PBA restored histone acetylation levels, increased motoneuron survival, life extent, and motor performances, as compared with vehicle-treated animals, and, importantly, was effective even when administered at disease onset [12,13]”.

  • In the section concerning miRNAs, although miRNAs predominantly work through the mechanism described by the authors, there do exist exceptions in which miRNA positively regulate the translation of the target mRNA, such as miR-10a (doi: 10.1016/j.molcel.2008.05.001). Thus, the first sentence needs to be rephrased. Furthermore, beside miR-206, miR-133b, miR-146a, please also discuss the recent work concerning miR-142a-5p/MFN1 axis (doi: 10.7150/thno.40857).

We agree with the Reviewer’s concern. We modified the sentence and introduced a new reference [14] (page 8, line 318 Ref.128 of the manuscript).  In addition, we discussed the work referred to the suggested reference. Page 8, lines 341-345 (Ref.144 of the manuscript):

“In a very recent paper by Yang X et al. it was demonstrated that in mice sciatic nerve transection model, miRNA-142a-5p functions as an important regulator of denervation-induced skeletal muscle atrophy by inducing mitochondrial dysfunction, mitophagy, and apoptosis by targeting membrane-anchored proteins, mitofusin-1 (MFN1) [15]”.

  • In the section concerning nutrition, it is worthwhile to mention certain diet supplement, such as sodium butyrate and phenylbutyrate (both are HDAC inhibitors), have been associated with the alleviation of ALS in animal models (doi: 10.1016/j.clinthera.2016.12.014; doi: 10.1016/j.nbd.2005.09.013)

We agree with the Reviewer’s concern and, in the new version of the manuscript, we discuss this topic. In particular, page 10, lines 421-434 (Ref.182 and 120) of the manuscript):

“In mice fed with sodium butyrate, a natural bacterial product, intestinal microbial homeostasis was restored, gut integrity was improved, and life span was prolonged compared with those of control mice. In both ALS mice and intestinal epithelial cell cultures derived from humans, sodium butyrate treatment was associated with decreased aggregation of the G93A SOD 1 mutated protein.  The findings from this study highlight the complex role of the gut microbiome and intestinal epithelium in the progression of ALS and indicate sodium butyrate as a potential therapeutic reagent for restoring ALS-related dysbiosis [16].

In addition, the histone deacetylase inhibitor PBA significantly improved motor function and extended survival in the G93A transgenic mouse model. This effect is enhanced when PBA is administrated in combination with the antioxidant AEOL 10150, suggesting that HDAC inhibitors and blocking oxidative stress agents may exert additive therapeutic effects in treating mutant-SOD1-associated ALS and other neurodegenerative diseases [17]”.

  • In the section concerning exercise, the impact of physiological Ca2+ transients on muscle ROS production could be included (doi: 10.1186/s13395-017-0123-0).

As suggested by the Reviewer, in the new version of the manuscript, we discussed this topic and added the new reference. In particular, pages 11 and 12, lines 456-466 (Ref.196 of the manuscript):

“Under the voltage-clamp condition, it has been found that the loss of physiological Ca2+ transients or mitochondrial Ca2+ uptake could be an initial trigger for mitochondrial dysfunction with increased mitochondrial ROS production in skeletal muscle fibers following denervation [18]. Both acute and long-term endurance exercises have been reported to activate certain signaling pathways to counteract ROS production. Meanwhile, electrical stimulation is known to help prevent apoptosis and alleviate muscle atrophy in denervated animal models and patients with motor impairment. Several studies focus on the excitation-transcription coupling framework to understand the beneficial role of exercise and electrical stimulation. Interestingly, a recent study has revealed an unexpected role of rapid mitochondrial Ca2+ transients in keeping mitochondrial permeability transition pore (mPTP) at a closed state with reduced mitochondrial ROS production” [18].

  • Page 2, line 67-69: “…DNA methylation, histone modification, and miRNAs-dependent gene expression represent the major epigenetic mechanisms that play a crucial role in NMJs remodeling [19-21]”. Those three cited references have no discussion on NMJ remodeling.

Thanks, we have replaced the references with the following (page 2 lines 85 and 86, Ref.46-48 of the manuscript):

Latcheva NK, Viveiros JM, Waddell EA, Nguyen PTT, Liebl FLW, Marenda DR. Epigenetic crosstalk: Pharmacological inhibition of HDACs can rescue defective synaptic morphology and neurotransmission phenotypes associated with loss of the chromatin reader Kismet. Mol Cell Neurosci. 2018 Mar;87:77-85. doi: 10.1016/j.mcn.2017.11.007. Epub 2017 Dec 15. PMID: 29249293 [19].

Osseni A, Ravel-Chapuis A, Thomas JL, Gache V, Schaeffer L, Jasmin BJ. HDAC6 regulates microtubule stability and clustering of AChRs at neuromuscular junctions. J Cell Biol. 2020 Aug 3;219(8):e201901099. doi: 10.1083/jcb.201901099. PMID: 32697819; PMCID: PMC7401804 [20].

Lu, L., Liu, Y., Liu, Y. et al. Secreted miRNAs in the tripartite neuromuscular junction. ExRNA 1, 35 (2019). https://doi.org/10.1186/s41544-019-0019-8 [21].

  • Page 6, line 244-245: "…overexpression of Sirt1 in the brain of aged mice correlates with a higher percentage of innervated NMJs, less fragmented AChR clusters, and in increased number of NMJs…[74,75]”. Again, those two cited references (original studies) have no data on NMJ.

Thanks, we have replaced the references with the following (Page 7, line 305, Ref.124 and 125 of the manuscript):

SIRT1 deacetylase in aging‐induced neuromuscular degeneration and amyotrophic lateral sclerosis Adrianna Z. Herskovits  Tegan A. Hunter  Nicholas Maxwell  Katherine Pereira  Charles A. Whittaker  Gregorio Valdez  Leonard P. Guarente [22]

Snyder-Warwick AK, Satoh A, Santosa KB, Imai SI, Jablonka-Shariff A. Hypothalamic Sirt1 protects terminal Schwann cells and neuromuscular junctions from age-related morphological changes. Aging Cell. 2018 Aug;17(4):e12776. doi: 10.1111/acel.12776. Epub 2018 May 30. PMID: 29851253; PMCID: PMC6052483.[23]

  • Page 6, line 259: a reference number should be included for Valdez G. et al

We introduced this reference (page 8, line 328 Ref.23 of the manuscript).

  • Page 6: Line 259-261: the citation 91 and 92 have no study on the formation of new NMJs following nerve injury.

We are sorry for this mistake, we removed these references.

  • Page 7: Line 317-320: “Another interesting study demonstrates that caloric restriction and resveratrol, which is a caloric restriction mimetic, can attenuate NMJ fragmentation and degradation partially through the Sirt1, peroxisome proliferator-activated receptor-gamma coactivator alpha (PGC1α), and mTOR signaling pathways [119–122].” None of those listed citations has studies on NMJ.  Additionally, when discussing the beneficial effect of resveratrol, the authors should also be aware and discuss the controversial experimental evidence on resveratrol, to avoid a potential overstatement and misleading readers.

Thanks for this observation. In the new version of the manuscript, we rephrased the sentence and added appropriated references (page 10, lines 433-437 Ref.23 and 183-189 of the manuscript). In particular:

“Another interesting study demonstrates that caloric restriction and resveratrol, a small natural polyphenol, which is a caloric restriction mimetic, can attenuate NMJ fragmentation and degradation in a mouse model of ALS [24,25]. In particular, it has been shown that resveratrol, when used at certain doses, is capable of slowing NMJs degeneration, while at other doses does not affect the disease progression [25–31]”.

  • Minor issues:
  • Please add in the legend of Fig. 2 that CAT is the abbreviation of catalase.

We added that CAT is the abbreviation of catalase (page 6 line 235).

  • In the section concerning exercise, PGC1α was misspelled as PCG1α twice.

We are sorry for the misspelled; we corrected it in the new version of the manuscript

References

  1. Li, A.; Yi, J.; Li, X.; Zhou, J. Physiological Ca2+ Transients Versus Pathological Steady-State Ca2+ Elevation, Who Flips the ROS Coin in Skeletal Muscle Mitochondria. Front. Physiol. 2020, 11.
  2. Tajrishi, M.M.; Shin, J.; Hetman, M.; Kumar, A. DNA methyltransferase 3a and mitogen-activated protein kinase signaling regulate the expression of fibroblast growth factor-inducible 14 (Fn14) during denervation-induced skeletal muscle atrophy. J. Biol. Chem. 2014, 289, 19985–19999.
  3. Mittal, A.; Bhatnagar, S.; Kumar, A.; Lach-Trifilieff, E.; Wauters, S.; Li, H.; Makonchuk, D.Y.; Glass, D.J.; Kumar, A. The TWEAK-Fn14 system is a critical regulator of denervation-induced skeletal muscle atrophy in mice. J. Cell Biol. 2010, 188, 833–849.
  4. Bhatnagar, S.; Kumar, A. The TWEAK-Fn14 System: Breaking the Silence of Cytokine-Induced Skeletal Muscle Wasting. Curr. Mol. Med. 2011, 12, 3–13.
  5. Rouaux, C.; Jokic, N.; Mbebi, C.; Boutillier, S.; Loeffler, J.P.; Boutillier, A.L. Critical loss of CBP/p300 histone acetylase activity by caspase-6 during neurodegeneration. EMBO J. 2003, 22, 6537–6549.
  6. Rouaux, C.; Loeffler, J.P.; Boutillier, A.L. Targeting CREB-binding protein (CBP) loss of function as a therapeutic strategy in neurological disorders. In Proceedings of the Biochemical Pharmacology; Elsevier, 2004; Vol. 68, pp. 1157–1164.
  7. Saha, R.N.; Pahan, K. HATs and HDACs in neurodegeneration: A tale of disconcerted acetylation homeostasis. Cell Death Differ. 2006, 13, 539–550.
  8. Langley, B.; Gensert, J.A.M.; Beal, M.F.; Ratan, R.R. Remodeling chromatin and stress resistance in the central nervous system: Histone deacetylase inhibitors as novel and broadly effective neuroprotective agents. Curr. Drug Targets CNS Neurol. Disord. 2005, 4, 41–50.
  9. Yoo, Y.E.; Ko, C.P. Treatment with trichostatin A initiated after disease onset delays disease progression and increases survival in a mouse model of amyotrophic lateral sclerosis. Exp. Neurol. 2011, 231, 147–159.
  10. Piepers, S.; Veldink, J.H.; De Jong, S.W.; Van Der Tweel, I.; Van Der Pol, W.L.; Uijtendaal, E. V.; Schelhaas, H.J.; Scheffer, H.; De Visser, M.; De Jong, J.M.B.V.; et al. Randomized sequential trial of valproic acid in amyotrophic lateral sclerosis. Ann. Neurol. 2009, 66, 227–234.
  11. Rouaux, C.; Panteleeva, I.; René, F.; De Aguilar, J.L.G.; Echaniz-Laguna, A.; Dupuis, L.; Menger, Y.; Boutillier, A.L.; Loeffler, J.P. Sodium valproate exerts neuroprotective effects in vivo through CREB-binding protein-dependent mechanisms but does not improve survival in an amyotrophic lateral sclerosis mouse model. J. Neurosci. 2007, 27, 5535–5545.
  12. Petri, S.; Kiaei, M.; Kipiani, K.; Chen, J.; Calingasan, N.Y.; Crow, J.P.; Beal, M.F. Additive neuroprotective effects of a histone deacetylase inhibitor and a catalytic antioxidant in a transgenic mouse model of amyotrophic lateral sclerosis. Neurobiol. Dis. 2006, 22, 40–49.
  13. Ryu, H.; Smith, K.; Camelo, S.I.; Carreras, I.; Lee, J.; Iglesias, A.H.; Dangond, F.; Cormier, K.A.; Cudkowicz, M.E.; Brown, R.H.; et al. Sodium phenylbutyrate prolongs survival and regulates expression of anti-apoptotic genes in transgenic amyotrophic lateral sclerosis mice. J. Neurochem. 2005, 93, 1087–1098.
  14. Ørom, U.A.; Nielsen, F.C.; Lund, A.H. MicroRNA-10a Binds the 5′UTR of Ribosomal Protein mRNAs and Enhances Their Translation. Mol. Cell 2008, 30, 460–471.
  15. Yang, X.; Xue, P.; Chen, H.; Yuan, M.; Kang, Y.; Duscher, D.; Machens, H.G.; Chen, Z. Denervation drives skeletal muscle atrophy and induces mitochondrial dysfunction, mitophagy and apoptosis via miR-142a-5p/MFN1 axis. Theranostics 2020, 10, 1415–1432.
  16. Zhang, Y. guo; Wu, S.; Yi, J.; Xia, Y.; Jin, D.; Zhou, J.; Sun, J. Target Intestinal Microbiota to Alleviate Disease Progression in Amyotrophic Lateral Sclerosis. Clin. Ther. 2017, 39, 322–336.
  17. Petri, S.; Kiaei, M.; Kipiani, K.; Chen, J.; Calingasan, N.Y.; Crow, J.P.; Beal, M.F. Additive neuroprotective effects of a histone deacetylase inhibitor and a catalytic antioxidant in a transgenic mouse model of amyotrophic lateral sclerosis. Neurobiol. Dis. 2006, 22, 40–49.
  18. Karam, C.; Yi, J.; Xiao, Y.; Dhakal, K.; Zhang, L.; Li, X.; Manno, C.; Xu, J.; Li, K.; Cheng, H.; et al. Absence of physiological Ca2+ transients is an initial trigger for mitochondrial dysfunction in skeletal muscle following denervation. Skelet. Muscle 2017, 7, 6.
  19. Latcheva, N.K.; Viveiros, J.M.; Waddell, E.A.; Nguyen, P.T.T.; Liebl, F.L.W.; Marenda, D.R. Epigenetic crosstalk: Pharmacological inhibition of HDACs can rescue defective synaptic morphology and neurotransmission phenotypes associated with loss of the chromatin reader Kismet. Mol. Cell. Neurosci. 2018, 87, 77–85.
  20. Osseni, A.; Ravel-Chapuis, A.; Thomas, J.-L.; Gache, V.; Schaeffer, L.; Jasmin, B.J. HDAC6 regulates microtubule stability and clustering of AChRs at neuromuscular junctions. J. Cell Biol. 2020, 219.
  21. Lu, L.; Liu, Y.; Liu, Y.; Zhang, F.; Wang, H.; Zhang, Q.; Pan, F. Secreted miRNAs in the tripartite neuromuscular junction. ExRNA 2019, 1, 1–4.
  22. Herskovits, A.Z.; Hunter, T.A.; Maxwell, N.; Pereira, K.; Whittaker, C.A.; Valdez, G.; Guarente, L.P. SIRT1 deacetylase in aging-induced neuromuscular degeneration and amyotrophic lateral sclerosis. Aging Cell 2018, 17.
  23. Snyder-Warwick, A.K.; Satoh, A.; Santosa, K.B.; Imai, S. ichiro; Jablonka-Shariff, A. Hypothalamic Sirt1 protects terminal Schwann cells and neuromuscular junctions from age-related morphological changes. Aging Cell 2018, 17.
  24. Valdez, G.; Tapia, J.C.; Kang, H.; Clemenson, G.D.; Gage, F.H.; Lichtman, J.W.; Sanes, J.R. Attenuation of age-related changes in mouse neuromuscular synapses by caloric restriction and exercise. Proc. Natl. Acad. Sci. U. S. A. 2010, 107, 14863–14868.
  25. Stockinger, J.; Maxwell, N.; Shapiro, D.; deCabo, R.; Valdez, G. Caloric Restriction Mimetics Slow Aging of Neuromuscular Synapses and Muscle Fibers. J. Gerontol. A. Biol. Sci. Med. Sci. 2017, 73, 21–28.
  26. Bar, S.; Prasad, M.; Datta, R. Neuromuscular degeneration and locomotor deficit in a Drosophila model of mucopolysaccharidosis VII is attenuated by treatment with resveratrol. DMM Dis. Model. Mech. 2018, 11.
  27. Han, S.; Choi, J.R.; Soon Shin, K.; Kang, S.J. Resveratrol upregulated heat shock proteins and extended the survival of G93A-SOD1 mice. Brain Res. 2012, 1483, 112–117.
  28. Kim, D.; Nguyen, M.D.; Dobbin, M.M.; Fischer, A.; Sananbenesi, F.; Rodgers, J.T.; Delalle, I.; Baur, J.A.; Sui, G.; Armour, S.M.; et al. SIRT1 deacetylase protects against neurodegeneration in models for Alzheimer’s disease and amyotrophic lateral sclerosis. EMBO J. 2007, 26, 3169–3179.
  29. Mancuso, R.; del Valle, J.; Modol, L.; Martinez, A.; Granado-Serrano, A.B.; Ramirez-Núñez, O.; Pallás, M.; Portero-Otin, M.; Osta, R.; Navarro, X. Resveratrol Improves Motoneuron Function and Extends Survival in SOD1G93A ALS Mice. Neurotherapeutics 2014, 11, 419–432.
  30. Markert, C.D.; Kim, E.; Gifondorwa, D.J.; Childers, M.K.; Milligan, C.E. A single-dose resveratrol treatment in a mouse model of amyotrophic lateral sclerosis. J. Med. Food 2010, 13, 1081–1085.
  31. Song, L.; Chen, L.; Zhang, X.; Li, J.; Le, W. Resveratrol ameliorates motor neuron degeneration and improves survival in SOD1G93A mouse model of amyotrophic lateral sclerosis. Biomed Res. Int. 2014, 2014.

Reviewer 3 Report

The review “Role of nutrition and exercise to counteract sarcopenia and age-related alterations at neuromuscular junction” presents a comprehensive overview of the most recent results on the importance of the neuromuscular junction integrity in maintaining muscle mass and performance.

In this context, the authors focus their attention on “muscle side”, rather than “motor neuron side” in discussing the involvement of alterations of neuromuscular junctions as a central feature of muscle dysfunction. Both oxidative stress and epigenetic pathways, whose alterations in skeletal muscle are known to bias neuromuscular junctions’ functionality ,are concisely but exhaustively presented. Accordingly, I suggest considering a change in the title of the review, pointing to oxidative stress and epigenetic control, rather than nutrition and exercise.

The review is clearly written, detailed and with an adequate number of references. No major questions can be raised.

Minor points:

Line 74: I would suggest replacing “composed” with “compose”

Lines 100-112: The role of the Agrin-Muscle-Specific Kinase-Low Density Lipoprotein receptor related protein 4 signalling appears to be very interesting in NMJ integrity. The last sentence of this paragraph is not very clear “…suggesting that agrin may cause denervation of NMJ”. I would suggest extending this point.

Line 114: please consider replacing “determines” with “determine”

Line 118: please add “of” after representation

Line 140: Amyotrophic Lateral Sclerosis was already abbreviated in line 63.               

The discussion of the role of d Sirt1 in regulating histone acetylation may be confounding for general readers (lines 243-249). The authors may consider to better explain this part.

Only some of the miRNAs listed in lines 253-255 are discussed in the subsequent paragraphs (lines 259-283), while the role of additional miRNAs not previously listed in lines 253-255 is presented (i.e. miR-146/234, lines 274-283). I suggest adding a table in which the miRNAs, their role in sarcopenia/age-related muscle loss, and relative refs are reported.

Line 214: the authors include miRNA among epigenetic regulations of NMJ dysfunction. Given that the target of miRNA regulation is mainly at the translational level, we would suggest discussing miRNA and myo-miRNA in a dedicated paragraph.

Author Response

Minor points:

  • Line 74: I would suggest replacing “composed” with “compose”

Thank you, we replaced the word (page 2 , line 90).

  • Lines 100-112: The role of the Agrin-Muscle-Specific Kinase-Low Density Lipoprotein receptor related protein 4 signalling appears to be very interesting in NMJ integrity. The last sentence of this paragraph is not very clear “…suggesting that agrin may cause denervation of NMJ”. I would suggest extending this point.

Maybe the confusion is because the correct sentence in the text is “….suggesting that aging may cause denervation of NMJ” and not “…suggesting that agrin may cause denervation of NMJ” as reported by the Reviewer. However, the meaning of the sentence is: since Musk and Lrp4 are increased in aged muscle and are, also, upregulated in denervated skeletal muscles of young mice it is an attempt to speculate that aging may cause denervation of NMJs.

  • Line 114: please consider replacing “determines” with “determine”

Than you, we replaced the word (page 3 , line 130).

  • Line 118: please add “of” after representation

We added “of” (page 4, line 134).

  • Line 140: Amyotrophic Lateral Sclerosis was already abbreviated in line 63.

Thanks, we now used only the abbreviation here.

  • The discussion of the role of Sirt1 in regulating histone acetylation may be confounding for general readers (lines 243-249). The authors may consider to better explain this part.

Thank you for this suggestion. In the revised version of our manuscript, we better clarified the role of Sirt1 in NMJs protection during aging. In particular, page 7 lines 301-309:

“Sirt1 is a nicotinamide adenine dinucleotide (NAD+)‐dependent deacetylase, involved in ALS and in other neurodegenerative diseases, where it plays a crucial role in attenuating damages caused by aging [1,2].  Overexpression of Sirt1 in the brain of aged mice correlates with a higher percentage of innervated NMJs, less fragmented AChR clusters, and an increased number of NMJs covered by tSCs [3,4]. Also, in a recent paper by A. Z. Herskovits et al., it has been demonstrated that Sirt1 expression decreases in the spinal-cord of wild-type mice during normal aging and the overexpression of Sirt1 in motor neurons partially protects NMJs from the deleterious effects of aging and ALS [37]”.

  • Only some of the miRNAs listed in lines 253-255 are discussed in the subsequent paragraphs (lines 259-283), while the role of additional miRNAs not previously listed in lines 253-255 is presented (i.e. miR-146/234, lines 274-283). I suggest adding a table in which the miRNAs, their role in sarcopenia/age-related muscle loss, and relative refs are reported.

Thank you for your suggestion. In the new version of the manuscript, we added a table in which the miRNAs, their role in sarcopenia/age-related muscle loss, and relative references are reported (page 9).

  • Line 214: the authors include miRNA among epigenetic regulations of NMJ dysfunction. Given that the target of miRNA regulation is mainly at the translational level, we would suggest discussing miRNA and myo-miRNA in a dedicated paragraph.

Thank you for your suggestion. In the new version of the manuscript, we included miRNAs in a dedicated paragraph. 

References

  1. Kim, D.; Nguyen, M.D.; Dobbin, M.M.; Fischer, A.; Sananbenesi, F.; Rodgers, J.T.; Delalle, I.; Baur, J.A.; Sui, G.; Armour, S.M.; et al. SIRT1 deacetylase protects against neurodegeneration in models for Alzheimer’s disease and amyotrophic lateral sclerosis. EMBO J. 2007, 26, 3169–3179.
  2. Watanabe, S.; Ageta-Ishihara, N.; Nagatsu, S.; Takao, K.; Komine, O.; Endo, F.; Miyakawa, T.; Misawa, H.; Takahashi, R.; Kinoshita, M.; et al. SIRT1 overexpression ameliorates a mouse model of SOD1-linked amyotrophic lateral sclerosis via HSF1/HSP70i chaperone system. Mol. Brain 2014, 7.
  3. Snyder-Warwick, A.K.; Satoh, A.; Santosa, K.B.; Imai, S. ichiro; Jablonka-Shariff, A. Hypothalamic Sirt1 protects terminal Schwann cells and neuromuscular junctions from age-related morphological changes. Aging Cell 2018, 17.
  4. Herskovits, A.Z.; Hunter, T.A.; Maxwell, N.; Pereira, K.; Whittaker, C.A.; Valdez, G.; Guarente, L.P. SIRT1 deacetylase in aging-induced neuromuscular degeneration and amyotrophic lateral sclerosis. Aging Cell 2018, 17.

Reviewer 4 Report

The manuscript by Dobrowolny et al. reviews urgent issues in human health, the etiology of sarcopenia and potential therapeutic strategies for it. Specifically, they deal with highly relevant factors involved in sarcopenia such as ROS, epigenetic regulation, nutrients, and exercise, focusing on their impacts on NMJ alterations. However, the manuscript misses reviews of some important topics and suffers from several inaccurate/unclear citations and statements, which must be addressed before publication.

Abstract - Introduction of exercise and nutrition is abrupt. Briefly highlight why they are important, as suggested in the title.

Section 4.1 – The authors provided a relatively substantial amount of reviews on ROS and NMJ with a dedicated figure. However, they did not review the potential role of nutrients on ROS-regulation of NMJ and sarcopenia. Given the title of this manuscript, the potential link between nutrients-ROS-NMJ-sarcopenia must be discussed.

Section 4.2 – Similar to the comment above, the authors did not review the potential role of exercise on ROS/epigenetic regulation of NMJ and sarcopenia. Any reports on the effects of exercise on ROS system and epigenetic regulation in young and aged muscles?

Figure 2 – Based on the manuscript, this reviewer is not sure about ROS inhibition of SOD/CAT. Why not in the opposite direction? The same is true for the arrow from ROS to mitochondrial dysfunctions and impairment of autophagic flux. Also, in the legend, “the reduction in fiber number” is not clearly supported by the literatures in this review. The lines 192-195 seem relevant, but the sentence is difficult to understand and rather gives an impression that exercise or muscle contraction is detrimental to the aged muscle, which is counterintuitive to the main theme of this manuscript. Also, it misses citations.

Figure 3 – The legend says, “Nutrients and diet compounds may affect DNA methylation”. However, evidence for this is not clearly described in this review.

Conclusions – This section does not provide conclusions. Also need to include future perspectives.

Citation issues

The following sentences need (more) citations.

  • line 36 (… as sarcopenia.), line 55 (… muscle size), line 65 (… ALS.), line 67 (… NMJ.), line 96 (… response), line 200-202 (additional citation recommended: PMID: 33406419), line 216 (… diseases.), line 222 (… silencing.), line 239 (… function.),

The following sentences need more relevant citations. Double-check and replace as appropriate.

  • line 43 (… generation.), lines 48-50 (it is … denervation.), line 94 (… Ach decrease.), line 112 (… NMJs.), line 225 (68,69 are not recent findings), lines 242 (is 73 correct?), 246 (… tSCs), line 253 (… proliferation.)

Clarification issues

  • line 61: how does “exon sprouting” deteriorates NMJ?
  • line 63-65: does this mean that understanding NMJ deterioration in aging may be crucial to develop therapeutic strategies for other motor neuron diseases?
  • line 107: impaired Agrin expression? Specify what "impaired" means.
  • line 185: “the activation of an NADPH oxidase (NOX2) complex” in pre-synaptic nerve?
  • line 187: elaborate for broader readers and beginners.
  • line 339: endocrine??

All figures need higher resolution.

Author Response

Reviewer 4

  • The manuscript by Dobrowolny et al. reviews urgent issues in human health, the etiology of sarcopenia and potential therapeutic strategies for it. Specifically, they deal with highly relevant factors involved in sarcopenia such as ROS, epigenetic regulation, nutrients, and exercise, focusing on their impacts on NMJ alterations. However, the manuscript misses reviews of some important topics and suffers from several inaccurate/unclear citations and statements, which must be addressed before publication.

  • Abstract - Introduction of exercise and nutrition is abrupt. Briefly highlight why they are important, as suggested in the title.

In agreement with the Reviewer request, we modified the abstract by adding the following sentence:

It is established that environmental and lifestyle factors, such as physical exercise and nutrition, that are susceptible to change during aging, can modulate epigenetic phenomena and attenuate the age-related NMJs changes”.

In addition, we have changed the title of the manuscript to a more appropriate one.

  • Section 4.1 – The authors provided a relatively substantial amount of reviews on ROS and NMJ with a dedicated figure. However, they did not review the potential role of nutrients on ROS-regulation of NMJ and sarcopenia. Given the title of this manuscript, the potential link between nutrients-ROS-NMJ-sarcopenia must be discussed.

Thank you for this consideration. In the new version of the manuscript, we introduced this topic. See page 10, lines 408-432. There are a number of natural compounds that have been demonstrated to have antioxidants activity [1–7]. Vitamin D is a fat-soluble vitamin present in many foods although it is primarily produced in the skin when it has been exposed to ultraviolet (UV) rays [8]. The reduction of muscle mass observed with advancing aging has been associated with decreased circulating vitamin D levels, leading to frailty in the elderly and frequent falls [9–13]. It has been reported that in a mouse model vitamin D deficiency leads to alteration in NMJ-related genes and protein expression levels [14]. Vitamin E of which many vegetables such as tomatoes, broccoli, and spinach are rich, has been demonstrated to prevent lipid peroxidation and to counteract the negative effects of free radicals in cellular membranes and lipoproteins revealing protective functions against neurodegenerative diseases [15]. In mice fed with sodium butyrate, a natural bacterial product, intestinal microbial homeostasis was restored, gut integrity was improved, and life span was prolonged compared with those of control mice. In both ALS mice and intestinal epithelial cell cultures derived from humans, sodium butyrate treatment was associated with decreased aggregation of the SOD1 mutated protein. The findings from this study highlight the complex role of the gut microbiome and intestinal epithelium in the progression of ALS and indicate sodium butyrate as a potential therapeutic reagent for restoring ALS-related dysbiosis [16]. In addition, the histone deacetylase inhibitor PBA significantly improved motor function and extended survival in the G93A transgenic mouse model. This effect is enhanced when PBA is administrated in combination with the antioxidant AEOL 10150, suggesting that HDAC inhibitors and blocking oxidative stress agents may exert additive therapeutic effects in treating mutant-SOD1-associated ALS and other neurodegenerative diseases [17]”.

  • Section 4.2 – Similar to the comment above, the authors did not review the potential role of exercise on ROS/epigenetic regulation of NMJ and sarcopenia. Any reports on the effects of exercise on ROS system and epigenetic regulation in young and aged muscles?

As required by this Reviewer, in the new version of the manuscript we introduced the following (pages 11 and 12, lines 456-466):

“Under the voltage-clamp condition, it has been found that the loss of physiological Ca2+ transients or mitochondrial Ca2+ uptake could be an initial trigger for mitochondrial dysfunction with increased mitochondrial ROS production in skeletal muscle fibers following denervation [18]. Both acute and long-term endurance exercises have been reported to activate certain signaling pathways to counteract ROS production. Meanwhile, electrical stimulation is known to help prevent apoptosis and alleviate muscle atrophy in denervated animal models and patients with motor impairment. Several studies focus on the excitation-transcription coupling framework to understand the beneficial role of exercise and electrical stimulation. Interestingly, a recent study has revealed an unexpected role of rapid mitochondrial Ca2+ transients in keeping mitochondrial permeability transition pore (mPTP) at a closed state with reduced mitochondrial ROS production”.

and (page 12, lines 471-480):

“As previously mentioned, ROS production during physical activity modules signaling pathways involved in muscle remodeling and in the adaptive response necessary to control oxidative stress [19]. Among the principal redox-sensitive signaling pathways that play a very important role in the skeletal muscle adaptive response to oxidative stress are Nuclear Factor kappa-light-chain-enhancer of activated B cells (NF-κB), Mitogen-Activated Protein Kinases (MAPKs), and PGC1α [20]. In particular, NF-kB induced by increased levels of hydrogen peroxide and/or inflammatory cytokines leads to high levels of molecules such as  iNOS, cyclooxygenase 2 (COX2), and SOD2 [21], while ROS-dependent activation of MAPK signaling results in increased glucose uptake by muscle fibers during exercise [22]”.

  • Figure 2 – Based on the manuscript, this reviewer is not sure about ROS inhibition of SOD/CAT. Why not in the opposite direction? The same is true for the arrow from ROS to mitochondrial dysfunctions and impairment of autophagic flux. Also, in the legend, “the reduction in fiber number” is not clearly supported by the literatures in this review. The lines 192-195 seem relevant, but the sentence is difficult to understand and rather gives an impression that exercise or muscle contraction is detrimental to the aged muscle, which is counterintuitive to the main theme of this manuscript. Also, it misses citations.

Thanks for these observations. We modified Figure 2 and replaced “the reduction in fiber number” with “ fiber type specification and innervation”. In addition, according to the Reviewer’s concern, we rephrased this part of the text. In particular, page 5, lines 213-217:

Studies in old muscles demonstrated an increased concentration of ROS during an acute bout of exercise, while, on the contrary, a protective effect against oxidative damage has been demonstrated by chronic exercise [19,23]. Moreover, physical inactivity induces a high level of oxidative stress leading to the onset of sarcopenia [24].

  • Figure 3 – The legend says, “Nutrients and diet compounds may affect DNA methylation”. However, evidence for this is not clearly described in this review.

We agree with the Reviewer concern, so we decided to modify Fig. 3 removing the effects of nutrients on DNA methylation.

  • Conclusions – This section does not provide conclusions. Also need to include future perspectives.

According to the Reviewer’s concern, we modified the conclusion section and we included future perspectives:

Conclusion and Future Perspectives

“This review highlights the recent epigenetic findings related to the NMJ dysregulation during aging and/or neuromuscular disorders. In addition, it discusses the role of physical activity and nutritional intervention as potential tools to ameliorate aging-associated decline or degenerative disease conditions by protecting and maintaining NMJ integrity. Future studies are needed to better clarify the molecular mechanisms underpinning lifestyle influence on NMJs homeostasis, in order to develop new therapeutic strategies to counteract the negative effect of aging in the neuromuscular system”.

  • Citation issues

The following sentences need (more) citations.

line 36 (… as sarcopenia.), line 55 (… muscle size), line 65 (… ALS.), line 67 (… NMJ.), line 96 (… response), line 200-202 (additional citation recommended: PMID: 33406419), line 216 (… diseases.), line 222 (… silencing.), line 239 (… function.),

As required, we added the following citations:

line 36 (… as sarcopenia.):

  • Wu R, De Vito G, Delahunt E, Ditroilo M. Age-related Changes in Motor Function (I). Mechanical and Neuromuscular Factors. Int J Sports Med. 2020 Oct;41(11):709-719. doi: 10.1055/a-1144-3408. Epub 2020 May 4. PMID: 32365388.
  • Larsson L, Degens H, Li M, Salviati L, Lee YI, Thompson W, Kirkland JL, Sandri M. Sarcopenia: Aging-Related Loss of Muscle Mass and Function. Physiol Rev. 2019 Jan 1;99(1):427-511. doi: 10.1152/physrev.00061.2017. PMID: 30427277; PMCID: PMC6442923.
  • Hepple RT, Rice CL. Innervation and neuromuscular control in ageing skeletal muscle. J Physiol. 2016 Apr 15;594(8):1965-78. doi: 10.1113/JP270561. Epub 2015 Dec 21. PMID: 26437581; PMCID: PMC4933121.

line 55 (… muscle size):

  • Rowan, S.L.; Rygiel, K.; Purves-Smith, F.M.; Solbak, N.M.; Turnbull, D.M.; Hepple, R.T. Denervation causes fiber atrophy and myosin heavy chain co-expression in senescent skeletal muscle. PLoS One 2012, 7, 29082.
  • Hepple RT, Rice CL. Innervation and neuromuscular control in ageing skeletal muscle. J Physiol. 2016 Apr 15;594(8):1965-78. doi: 10.1113/JP270561. Epub 2015 Dec 21. PMID: 26437581; PMCID: PMC4933121.

line 65 (… ALS.):

  • Pratt, J.; De Vito, G.; Narici, M.; Boreham, C. Neuromuscular Junction Aging: A Role for Biomarkers and Exercise. J. Gerontol. A. Biol. Sci. Med. Sci. 2021, 76, 576–585.
  • Lepore, E.; Casola, I.; Dobrowolny, G.; Musarò, A. Neuromuscular Junction as an Entity of Nerve-Muscle Communication. Cells 2019, 8, 906.

line 67 (… NMJ.):

  • Coppedè F. Epigenetics of neuromuscular disorders. Epigenomics. 2020 Dec;12(23):2125-2139. doi: 10.2217/epi-2020-0282. Epub 2020 Nov 6. PMID: 33155830.
  • Bennett, S.A.; Tanaz, R.; Cobos, S.N.; Torrente, M.P. Epigenetics in amyotrophic lateral sclerosis: a role for histone post-translational modifications in neurodegenerative disease. Transl. Res. 2019, 204, 19–30.
  • Azpurua, J.; Eaton, B.A. Neuronal epigenetics and the aging synapse. Front. Cell. Neurosci. 2015, 9.

line 96 (… response):

  • Taetzsch, T.; Valdez, G. NMJ maintenance and repair in aging. Curr. Opin. Physiol. 2018, 4, 57–64.

line 200-202 (additional citation recommended: PMID: 33406419):

  • You JS, Singh N, Reyes-Ordonez A, Khanna N, Bao Z, Zhao H, Chen J. ARHGEF3 Regulates Skeletal Muscle Regeneration and Strength through Autophagy. Cell Rep. 2021 Jan 5;34(1):108594. doi: 10.1016/j.celrep.2020.108594. Erratum in: Cell Rep. 2021 Feb 9;34(6):108731. PMID: 33406419.

line 216 (… diseases.):

  • Coppedè F. Epigenetics of neuromuscular disorders. Epigenomics. 2020 Dec;12(23):2125-2139. doi: 10.2217/epi-2020-0282. Epub 2020 Nov 6. PMID: 33155830.
  • Adam P. Sharples, Robert A. Seaborne, Claire E. Stewart, Chapter 19 - Epigenetics of Skeletal Muscle Aging, Editor(s): Alexey Moskalev, Alexander M. Vaiserman, In Translational Epigenetics, Epigenetics of Aging and Longevity, Academic Press,Volume 4, 2018, Pages 389-416, ISSN 25425358, ISBN 9780128110607
  • Gensous, N.; Bacalini, M.G.; Pirazzini, C.; Marasco, E.; Giuliani, C.; Ravaioli, F.; Mengozzi, G.; Bertarelli, C.; Palmas, M.G.; Franceschi, C.; et al. The epigenetic landscape of age-related diseases: the geroscience Biogerontology 2017, 18, 549–559.

line 222 (… silencing.):

Coppedè F. Epigenetics of neuromuscular disorders. Epigenomics. 2020 Dec;12(23):2125-2139. doi: 10.2217/epi-2020-0282. Epub 2020 Nov 6. PMID: 33155830.

line 239 (… function.)

Pigna, E.; Simonazzi, E.; Sanna, K.; Bernadzki, K.M.; Proszynski, T.; Heil, C.; Palacios, D.; Adamo, S.; Moresi, V. Histone deacetylase 4 protects from denervation and skeletal muscle atrophy in a murine model of amyotrophic lateral sclerosis. EBioMedicine 2019, 40, 717–732.

  • The following sentences need more relevant citations. Double-check and replace as appropriate.

line 43 (… generation.), lines 48-50 (it is … denervation.), line 94 (… Ach decrease.), line 112 (… NMJs.), line 225 (68,69 are not recent findings), lines 242 (is 73 correct?), 246 (… tSCs), line 253 (… proliferation.)

As required, we added the following citations:

line 43 (… generation.):

all the literature refers to the cited paper

lines 48-50 (it is … denervation.):

Bao, Z.; Cui, C.; Chow, S.K.H.; Qin, L.; Wong, R.M.Y.; Cheung, W.H. AChRs Degeneration at NMJ in Aging-Associated Sarcopenia–A Systematic Review. Front. Aging Neurosci. 2020, 12, 597811.

line 94 (… Ach decrease.):

Bowen DC, Park JS, Bodine S, Stark JL, Valenzuela DM, Stitt TN, Yancopoulos GD, Lindsay RM, Glass DJ, DiStefano PS. Localization and regulation of MuSK at the neuromuscular junction. Dev Biol. 1998 Jul 15;199(2):309-19. doi: 10.1006/dbio.1998.8936. PMID: 9698449.

line 225 (68,69 are not recent findings):

we have replaced “recent” with “several”

lines 242 (is 73 correct?):

we replaced reference 73 with the following:

Pigna, E.; Simonazzi, E.; Sanna, K.; Bernadzki, K.M.; Proszynski, T.; Heil, C.; Palacios, D.; Adamo, S.; Moresi, V. Histone deacetylase 4 protects from denervation and skeletal muscle atrophy in a murine model of amyotrophic lateral sclerosis. EBioMedicine 2019, 40, 717–732.

  • Clarification issues

line 61: how does “exon sprouting” deteriorates NMJ?

line 63-65: does this mean that understanding NMJ deterioration in aging may be crucial to develop therapeutic strategies for other motor neuron diseases?

line 107: impaired Agrin expression? Specify what "impaired" means.

line 185: “the activation of an NADPH oxidase (NOX2) complex” in pre-synaptic nerve?

line 187: elaborate for broader readers and beginners.

line 339: endocrine??

All figures need higher resolution.

We thank the reviewer and we clarify all the issues that he/she requested:

  • line 61: how does “exon sprouting” deteriorates NMJ?

In line 61 we refer to aging-related alteration of NMJ in terms of "...multiple innervations and axon sprouting". Indeed, in the work of Valdez and coauthors [25] it has been demonstrated that "..aged muscle exhibit some NMJs with two axons converged at the same postsynaptic site...". The author observed sprouting of axons to a NMJ that was partially or completely denervated and thin sprouts that extended beyond a postsynaptic site". 

  • line 63-65: does this mean that understanding NMJ deterioration in aging may be crucial to develop therapeutic strategies for other motor neuron diseases?

We believe that understanding the molecular mechanism of NMJ remodeling during aging will help to develop new strategies to attenuate muscle-nerve communication defects associated with aging.

  • line 107: impaired Agrin expression? Specify what "impaired" means.

Here we refer to a reduced level of agrin at the NMJs that leads to an impairment of agrin NMJ function. Indeed excessive agrin cleavage following neurotrypsin overexpression in motoneurons of transgenic mice results in severe NMJ alterations and histopathological alterations mimicking those of sarcopenia. In the revised version of the article we change "impaired" with "..reduced level of agrin at the NMJs..."

  • line 185: “the activation of an NADPH oxidase (NOX2) complex” in pre-synaptic nerve?
  • line 187: elaborate for broader readers and beginners

Here we refer to the activation of an NADPH oxidase (NOX2) complex during contractile activity of young rodent muscles. Muscle contractions leads to extracellular release of ATP into the NMJ synaptic cleft and induces the assembly of the NADPH oxidase complex in muscle tissue. ATP and NOX2 regulate acetylcholine (ACh) release from the pre-synaptic terminal. Therefore, in the revised version of the manuscript we better clarify these points and we change in: ".... Contractile activity in muscles of young rodents leads to the extracellular release of ATP and the activation of an NADPH oxidase (NOX2) complex in muscle post-synaptic terminal. NOX2 is responsible for superoxide and hydrogen peroxide (H2O2) production and regulates, together with ATP, the release of ACh from the motor nerve terminal”.

  • line 339: endocrine??

In 1961 Goldstein postulated, by cross-transfusion experiments, the existence of humoral components induced by muscular work [26] (Goldstein 1961) and for several  years the researchers were focused on elucidating the mechanisms by which skeletal muscle exercise can impact the entire body to promote profound health benefits [27]. More recently skeletal muscle has been defined as an endocrine organ, secreting a variety of factors, which are also called myokines [28]. Through autocrine, paracrine, and endocrine processes, myokines play different roles in skeletal muscle and in other organs and tissues, including the brain, liver, adipose tissue and immune cells [29]. Abnormalities in these factors may trigger and promote the pathogenesis of age-related and metabolic diseases, including sarcopenia.

References

  1. Monroy, A.; Lithgow, G.J.; Alavez, S. Curcumin and neurodegenerative diseases. BioFactors 2013, 39, 122–132.
  2. Pellavio, G.; Rui, M.; Caliogna, L.; Martino, E.; Gastaldi, G.; Collina, S.; Laforenza, U. Regulation of aquaporin functional properties mediated by the antioxidant effects of natural compounds. Int. J. Mol. Sci. 2017, 18.
  3. Boots, A.W.; Haenen, G.R.M.M.; Bast, A. Health effects of quercetin: From antioxidant to nutraceutical. Eur. J. Pharmacol. 2008, 585, 325–337.
  4. Wu, Q.; Wang, X.; Nepovimova, E.; Wang, Y.; Yang, H.; Li, L.; Zhang, X.; Kuca, K. Antioxidant agents against trichothecenes: New hints for oxidative stress treatment. Oncotarget 2017, 8, 110708–110726.
  5. Tresserra-Rimbau, A.; Arranz, S.; Vallverdu-Queralt, A. New Insights into the Benefits of Polyphenols in Chronic Diseases. Oxid. Med. Cell. Longev. 2017, 2017.
  6. Nieman, D.C.; Laupheimer, M.W.; Ranchordas, M.K.; Burke, L.M.; Stear, S.J.; Castell, L.M. A-Z of nutritional supplements: Dietary supplements, sports nutrition foods and ergogenic aids for health and performance - Part 33. Br. J. Sports Med. 2012, 46, 618–620.
  7. Damiano, S.; Sasso, A.; De Felice, B.; Di Gregorio, I.; La Rosa, G.; Lupoli, G.A.; Belfiore, A.; Mondola, P.; Santillo, M. Quercetin increases MUC2 and MUC5AC gene expression and secretion in intestinal goblet cell-like LS174T via PLC/PKCα/ERK1-2 pathway. Front. Physiol. 2018, 9.
  8. Uchitomi, R.; Oyabu, M.; Kamei, Y. Vitamin d and sarcopenia: Potential of vitamin D supplementation in sarcopenia prevention and treatment. Nutrients 2020, 12, 1–12.
  9. Cummings, S.R.; Kiel, D.P.; Black, D.M. Vitamin D Supplementation and increased risk of falling: A cautionary tale of vitamin supplements retold. JAMA Intern. Med. 2016, 176, 171–172.
  10. Kim, M.K.; Baek, K.H.; Song, K.H.; Il Kang, M.; Park, C.Y.; Lee, W.Y.; Won Oh, K. Vitamin D deficiency is associated with sarcopenia in older Koreans, regardless of obesity: The fourth Korea National Health and Nutrition Examination Surveys (KNHANES IV) 2009. J. Clin. Endocrinol. Metab. 2011, 96, 3250–3256.
  11. Huo, Y.R.; Suriyaarachchi, P.; Gomez, F.; Curcio, C.L.; Boersma, D.; Muir, S.W.; Montero-Odasso, M.; Gunawardene, P.; Demontiero, O.; Duque, G. Phenotype of Osteosarcopenia in Older Individuals With a History of Falling. J. Am. Med. Dir. Assoc. 2015, 16, 290–295.
  12. Snijder, M.B.; Van Schoor, N.M.; Pluijm, S.M.F.; Van Dam, R.M.; Visser, M.; Lips, P. Vitamin D status in relation to one-year risk of recurrent falling in older men and women. J. Clin. Endocrinol. Metab. 2006, 91, 2980–2985.
  13. Wintermeyer, E.; Ihle, C.; Ehnert, S.; Stöckle, U.; Ochs, G.; de Zwart, P.; Flesch, I.; Bahrs, C.; Nussler, A.K. Crucial role of vitamin D in the musculoskeletal system. Nutrients 2016, 8.
  14. Gifondorwa, D.J.; Thompson, T.D.; Wiley, J.; Culver, A.E.; Shetler, P.K.; Rocha, G. V.; Ma, Y.L.; Krishnan, V.; Bryant, H.U. Vitamin D and/or calcium deficient diets may differentially affect muscle fiber neuromuscular junction innervation. Muscle and Nerve 2016, 54, 1120–1132.
  15. Szymańska, R.; Nowicka, B.; Kruk, J. Vitamin E - Occurrence, Biosynthesis by Plants and Functions in Human Nutrition. Mini-Reviews Med. Chem. 2016, 17, 1039–1052.
  16. Zhang, Y. guo; Wu, S.; Yi, J.; Xia, Y.; Jin, D.; Zhou, J.; Sun, J. Target Intestinal Microbiota to Alleviate Disease Progression in Amyotrophic Lateral Sclerosis. Clin. Ther. 2017, 39, 322–336.
  17. Petri, S.; Kiaei, M.; Kipiani, K.; Chen, J.; Calingasan, N.Y.; Crow, J.P.; Beal, M.F. Additive neuroprotective effects of a histone deacetylase inhibitor and a catalytic antioxidant in a transgenic mouse model of amyotrophic lateral sclerosis. Neurobiol. Dis. 2006, 22, 40–49.
  18. Karam, C.; Yi, J.; Xiao, Y.; Dhakal, K.; Zhang, L.; Li, X.; Manno, C.; Xu, J.; Li, K.; Cheng, H.; et al. Absence of physiological Ca2+ transients is an initial trigger for mitochondrial dysfunction in skeletal muscle following denervation. Skelet. Muscle 2017, 7, 6.
  19. Damiano, S.; Muscariello, E.; La Rosa, G.; Di Maro, M.; Mondola, P.; Santillo, M. Dual role of reactive oxygen species in muscle function: Can antioxidant dietary supplements counteract age-related sarcopenia? Int. J. Mol. Sci. 2019, 20.
  20. Ji, L.L.; Zhang, Y. Antioxidant and anti-inflammatory effects of exercise: Role of redox signaling. Free Radic. Res. 2014, 48, 3–11.
  21. Ghosh, S.; Karin, M. Missing pieces in the NF-κB puzzle. Cell 2002, 109.
  22. Chambers, M.A.; Moylan, J.S.; Smith, J.D.; Goodyear, L.J.; Reid, M.B. Stretch-stimulated glucose uptake in skeletal muscle is mediated by reactive oxygen species and p38 MAP-kinase. J. Physiol. 2009, 587, 3363–3373.
  23. Bejma, J.; Ji, L.L. rapid communication Aging and acute exercise enhance free radical generation in rat skeletal muscle; 1999; Vol. 87;.
  24. Derbré, F.; Gratas-Delamarche, A.; Gómez-Cabrera, M.C.; Viña, J. Inactivity-induced oxidative stress: A central role in age-related sarcopenia? Eur. J. Sport Sci. 2014, 14.
  25. Valdez, G.; Tapia, J.C.; Kang, H.; Clemenson, G.D.; Gage, F.H.; Lichtman, J.W.; Sanes, J.R. Attenuation of age-related changes in mouse neuromuscular synapses by caloric restriction and exercise. Proc. Natl. Acad. Sci. U. S. A. 2010, 107, 14863–14868.
  26. GOLDSTEIN, M.S. Humoral nature of the hypoglycemic factor of muscular work. Diabetes 1961, 10, 232–234.
  27. Hoffmann, C.; Weigert, C. Skeletal muscle as an endocrine organ: The role of myokines in exercise adaptations. Cold Spring Harb. Perspect. Med. 2017, 7.
  28. Pedersen, B.K.; Febbraio, M.A. Muscles, exercise and obesity: Skeletal muscle as a secretory organ. Nat. Rev. Endocrinol. 2012, 8, 457–465.
  29. Weigert, C.; Lehmann, R.; Hartwig, S.; Lehr, S. The secretome of the working human skeletal muscle-A promising opportunity to combat the metabolic disaster? Proteomics - Clin. Appl. 2014, 8, 5–18.

Round 2

Reviewer 4 Report

The revision addressed most of the issues raised except for the following.

Thanks for these observations. We modified Figure 2 and replaced “the reduction in fiber number” with “ fiber type specification and innervation”. In addition, according to the Reviewer’s concern, we rephrased this part of the text. In particular, page 5, lines 213-217: Studies in old muscles demonstrated an increased concentration of ROS during an acute bout of exercise, while, on the contrary, a protective effect against oxidative damage has been demonstrated by chronic exercise [19,23]. Moreover, physical inactivity induces a high level of oxidative stress leading to the onset of sarcopenia [24].

  • This manuscript still does not seem to provide evidence/reference to support “Excessive ROS production during aging or degenerative disorder is responsible for the fiber type specification”. Please address this issue. Also, the rephrased part (213-217) is still unclear as to what message the authors want to convey to readers. Do the authors want to say that although an increased ROS during an acute bout (not but) of exercise may damage old muscles more than young muscles, repeated bouts of exercise/chronic exercise protects old muscles against oxidative damage (by what?)? CAT and SOD are no longer included in this figure and should be removed.

In 1961 Goldstein postulated, by cross-transfusion experiments, the existence of humoral components induced by muscular work [26] (Goldstein 1961) and for several years the researchers were focused on elucidating the mechanisms by which skeletal muscle exercise can impact the entire body to promote profound health benefits [27]. More recently skeletal muscle has been defined as an endocrine organ, secreting a variety of factors, which are also called myokines [28]. Through autocrine, paracrine, and endocrine processes, myokines play different roles in skeletal muscle and in other organs and tissues, including the brain, liver, adipose tissue and immune cells [29]. Abnormalities in these factors may trigger and promote the pathogenesis of age-related and metabolic diseases, including sarcopenia.

- This reviewer meant that the sentence only mentions autocrine and paracrine. Endocrine should also be included because the sentence says “… function on different argans…”.

Author Response

Reviewer 4

The revision addressed most of the issues raised except for the following.

Thanks for these observations. We modified Figure 2 and replaced “the reduction in fiber number” with “ fiber type specification and innervation”. In addition, according to the Reviewer’s concern, we rephrased this part of the text. In particular, page 5, lines 213-217: Studies in old muscles demonstrated an increased concentration of ROS during an acute bout of exercise, while, on the contrary, a protective effect against oxidative damage has been demonstrated by chronic exercise [19,23]. Moreover, physical inactivity induces a high level of oxidative stress leading to the onset of sarcopenia [24].

  • This manuscript still does not seem to provide evidence/reference to support “Excessive ROS production during aging or degenerative disorder is responsible for the fiber type specification”. Please address this issue.

According to the Reviewer's concern, we modified Figure 2. In particular, we changed “Excessive ROS production during aging or degenerative disorder is responsible for the fiber type specification”  with: “Excessive ROS production during aging or neurodegenerative disorder is responsible for the increased denervation and skeletal muscle atrophy”

  • Damiano S, Muscariello E, La Rosa G, Di Maro M, Mondola P, Santillo M. Dual Role of Reactive Oxygen Species in Muscle Function: Can Antioxidant Dietary Supplements Counteract Age-Related Sarcopenia? Int J Mol Sci. 2019 Aug 5;20(15):3815. doi: 10.3390/ijms20153815. PMID: 31387214; PMCID: PMC6696113.
  • Muller, F.L.; Song, W.; Liu, Y.; Chaudhuri, A.; Pieke-Dahl, S.; Strong, R.; Huang, T.T.; Epstein, C.J.; Roberts, L.J.; Csete, M.; et al. Absence of CuZn superoxide dismutase leads to elevated oxidative stress and acceleration of age-dependent skeletal muscle atrophy. Free Radic. Biol. Med. 2006, 40, 1993–2004.
  • Muller, F.L.; Song, W.; Jang, Y.C.; Liu, Y.; Sabia, M.; Richardson, A.; Van Remmen, H. Denervation-induced skeletal muscle atrophy is associated with increased mitochondrial ROS production. Am. J. Physiol. Regul. Integr. Comp. Physiol. 2007, 293, R1159–R1168.

  • Also, the rephrased part (213-217) is still unclear as to what message the authors want to convey to readers. Do the authors want to say that although an increased ROS during an acute bout (not but) of exercise may damage old muscles more than young muscles, repeated bouts of exercise/chronic exercise protects old muscles against oxidative damage (by what?)?

Thanks for this observation that allows us to better clarify this point. The sentence is replaced by:

It has been reported that exercise, through the activation of MAPK/NF-kB signaling pathways, can cause the expression of antioxidant enzymes which play a crucial role in the protection against ROS, and adaptation to exercise (1,2). However, the beneficial effects of physical activity are lost with exhaustive endurance and resistance exercise since the increased levels of ROS observed in these conditions overwhelm cellular antioxidants defenses, leading to tissue damage (3-5).

  • Gomez-Cabrera MC, Domenech E, Viña J. Moderate exercise is an antioxidant: upregulation of antioxidant genes by training. Free Radic Biol Med 2008; 44:126–131.
  • Wright VP, Reiser PJ, Clanton TL. Redox modulation of global phosphatase activity and protein phosphorylation in intact skeletal muscle. J Physiol 2009; 587 (Pt 23):5767–5781.
  • Sinha, S.; Ray, U.S.; Saha, M.; Singh, S.N.; Tomar, O.S. Antioxidant and redox status after maximal aerobic exercise at high altitude in acclimatized lowlanders and native highlanders. Eur. J. Appl. Physiol. 2009, 106, 807–814.
  • Sinha, S.; Singh, S.N.; Saha, M.; Kain, T.C.; Tyagi, A.K.; Ray, U.S. Antioxidant and oxidative stress responses of sojourners at high altitude in different climatic temperatures. Int. J. Biometeorol. 2010, 54, 85–92.
  • Bejma, J.; Ji, L.L. rapid communication Aging and acute exercise enhance free radical generation in rat skeletal muscle; 1999; Vol. 87.

CAT and SOD are no longer included in this figure and should be removed.

Sorry for the mistake. We remove CAT and SOD from the Figure legend.

In 1961 Goldstein postulated, by cross-transfusion experiments, the existence of humoral components induced by muscular work [26] (Goldstein 1961) and for several years the researchers were focused on elucidating the mechanisms by which skeletal muscle exercise can impact the entire body to promote profound health benefits [27]. More recently skeletal muscle has been defined as an endocrine organ, secreting a variety of factors, which are also called myokines [28]. Through autocrine, paracrine, and endocrine processes, myokines play different roles in skeletal muscle and in other organs and tissues, including the brain, liver, adipose tissue, and immune cells [29]. Abnormalities in these factors may trigger and promote the pathogenesis of age-related and metabolic diseases, including sarcopenia.

- This reviewer meant that the sentence only mentions autocrine and paracrine. Endocrine should also be included because the sentence says “… function on different argans…”.

The authors agree with the Reviewer’s concern. Page 12 line 469 we included “endocrine”.